

# Air Control Toolbox (ACT_v1.0): a machine learning flexible surrogate model to explore mitigation scenarios in air quality forecasts.

Augustin Colette[1], Laurence Rouïl[1], Frédérik Meleux[1], Vincent Lemaire[1,2], Blandine Raux[1]

[1] Institut National de l'Environnement Industriel et des Risques (INERIS), Parc Alata, BP2, 60550 Verneuil-en-Halatte, France

[2] now at Amplisim, 96b Boulevard Raspail, 75006 Paris, France

*Correspondence to*: Augustin Colette (augustin.colette@ineris.fr)

**Abstract.** We introduce the first toolbox that allows exploring the benefit of air pollution mitigation scenarios in the every-day air quality forecasts through a web interface. Chemistry-transport models (CTMs) are required to forecast air pollution episodes and assess the benefit that shall be expected from mitigation strategies. However, their complexity prohibits offering a high level of flexibility. The Air Control Toolbox relies on machine learning methods to cope with this limitation. It consists of a surrogate model trained on a limited set of sensitivity scenarios

to allow exploring any combination of mitigation measures. As such we take the best of the physical and chemical complexity of CTMs, operated on high performance computers for the everyday forecast, but we approximate a simplified response function that can be operated through a website to emulate the main sensitivities of the atmospheric system for a given day and location.

The numerical experimental plan to design the structure of the surrogate model is detailed by increasing level of

complexity. The selected structure of the surrogate is a quadrivariate polynomial of first order for residential heating emissions, and second order for agriculture, industry and traffic emissions with three interaction terms. It is fitted to 12 sensitivity CTM simulations, at each grid point and every day for $PM_{10}$, $PM_{2.5}$, $O_3$ (both as daily mean and daily maximum) and $NO_2$. The validation study demonstrates that we can keep relative errors below 2% at 95% of the grid points and days for all pollutants. Various applications of the toolbox are presented for air

quality episode analysis, source apportionment, and chemical regime analysis.



# 1    Introduction

The two most widespread applications of atmospheric chemistry modelling are (i) short term air quality forecasting, and (ii) long term analysis of mitigation strategies. We introduce here the first toolbox able to address both issues at once, so that the user can explore the benefit of any emission reduction control strategy for the current-day air quality forecast. Such a flexibility is provided by making the toolbox available through a web interface. The quality is ensured by relying on an emulator, or surrogate model, of a comprehensive air quality model.

The development of atmospheric chemistry transport models is motivated by the need to account for the dispersion and chemical evolution of pollutants in the atmosphere. A given influx of air pollutant emissions shall result in very different air concentrations depending on the meteorological conditions because of the dynamical and physical conditions (including advection, deposition, scavenging, turbulent mixing, …) and because of the chemical production and loss of secondary pollutants. As a result of this complexity of atmospheric chemistry and physics, there is no direct relationship between an incremental change of the emissions flux and resulting atmospheric concentrations.

Various approaches to air quality forecasting were developed since the 1970s (Zhang et al., 2012). The first approaches relied on statistical regressions between observed air pollutant concentrations and various precursors (mainly meteorological variables). But such approaches faced structural limitations, chiefly in accounting for non-local air pollution brought about by long range transport. So that 3D chemistry transport models are now widespread, for instance in the European Air quality forecasts operated by the Copernicus Atmosphere Monitoring Services (Marécal et al., 2015;Engelen and Peuch, 2017).

Three-dimensional chemistry transport models are also generally used to assess the benefit expected from a given air pollution control strategy (e.g. (Colette et al., 2012)). In that case, the computational burden increases because annual or multi-annual simulations are required. Such a burden constitutes a substantial limitation for the assessment of air pollution control strategies, where the end user is generally interested to compare the relative benefit of a series on individual mitigation measures. The replication of long-term chemistry transport simulation for each mitigation strategy becomes then prohibitive.

That is why alternate techniques were developed for decision support applications by means of surrogate models that consist in simplified regressions fitted to the response of a comprehensive air quality model (Cohan and Napelenok, 2011;Amann et al., 2011;Pisoni et al., 2017). Such surrogate models generally rely on the assumption





that the response to an incremental emission change can be approximated with a linear fit over a limited range of emission reduction magnitude. Such an approximation is required to limit the degrees of freedom of the surrogate model, especially when some geographical variability is accounted for in the surrogate model (i.e. when the emission reduction may vary in space, when exploring different magnitude of emission reduction by country for

instance). The linearity assumption is also supported by limiting the scope of surrogate models to long term indicators, such as annual mean exposure, which have a more linear response than short term responses (Thunis et al., 2015).

In the present paper, we introduce a first surrogate response model (denoted the Air Control Toolbox, ACT https://policy.atmosphere.copernicus.eu/CAMS_ACT.php) able to capture the non-linear response of any

magnitude of emission reduction for every-day air quality forecast. The model is designed to capture the daily means of both the $PM_{10}$ and $PM_{2.5}$ fractions of particulate pollution and nitrogen dioxide ($NO_2$) as well as the daily mean and daily maximum of ozone ($O_3avg$, $O_3max$). The spatial coverage is the greater European continent.

To offer the required level of flexibility for any user to explore their own mitigation scenario through a web interface, the model is simplified down to the level of a polynomial function. Nevertheless, all the complexity of

atmospheric chemistry is embedded in this surrogate which is fitted every day based on a subset (typically 10 to 15) of complete chemistry-transport model simulations, thereby relying on automated machine learning approach. As a result, the surrogate polynomial model accounts for all the important processes bearing upon the forecasted air quality, including long-range transport and chemistry. The only two simplifications limiting the range of application of ACT are that emission reductions are assumed to apply (i) over the long term (so that it is not

possible to investigate an emergency mitigation measures) and (ii) over the whole modelling domain uniformly.

Once the polynomial structure of the model is decided, an important part of the work consist in identifying the optimal set of those 10 to 15 chemistry-transport simulations that allow fitting a surrogate with adequate performances over any air quality situation. Most of the present paper is devoted to the description on how this numerical experiment plan is developed. The various methods (including overall surrogate structure, input data

and the underlying air quality forecast system) are presented in Section 2, the step-by-step development and validation of the surrogate ACT model is introduced in Section 3, in Section 4 we present the web interface as well as a few case studies, and the use of the model in source allocation and chemical regime analysis is discussed in Section 5 and 6, respectively.





## 2 Methods

### 2.1 Testing periods

The ambition of the ACT surrogate model is to apply to the every-day air quality forecast. The surrogate model should therefore present satisfactory performances in any situation, but especially during high air pollution
episodes. For the development and validation purposes of the present article we identified three case studies that are selected for their variety in terms of air pollution situation:

- a case of intense cold spell with typical wintertime particulate matter pollution (December 2016 - January 2017, (Forêt et al., 2017))
- a springtime PM episode dominated by ammonium nitrate formation (March 2015, (Petit et al., 2017))
- a summertime ozone episode (June 2017, (Tarrason et al., 2017)).

Note however that those periods do not constitute a specific training period for the surrogate model which is intended to be re-fitted to new CTM simulations every forecasted day in an automated machine learning approach. Those episodes are only selected to identify the best design for the surrogate and assess its performances in reproducing the full CTM.

### 2.2 Chemistry-Transport Model

The air quality simulations used for both the design of the numerical experiment and the every-day training of the ACT tool are performed with the CHIMERE Chemistry-Transport Model (Mailler et al., 2017;Menut et al., 2013). The model is widely used for air quality research and application ranging from short term forecasting (Marécal et al., 2015) to projection at climate scale (Colette et al., 2015).

We use a simulation setup similar to the operational regional forecast performed under the Copernicus Atmosphere Monitoring Service[1], albeit with a lower spatial resolution: 0.25 degree instead of 0.1 degree. The chimere model version is chimere2016a using Melchior gas phase chemistry, a two-product organic aerosol scheme and ISORROPIA thermodynamics.

The anthropogenic emissions in the reference simulations are TNO-MACCIII (Kuenen et al., 2011).
Meteorological data are operational analyses of the IFS (integrated forecasting system) model of the European

---

[1] http://regional.atmosphere.copernicus.eu





Centre for Medium Range Weather Forecasts. The chemical boundary conditions are also obtained from ECMWF, also with the IFS model.

### 2.3 Surrogate model structure

The structure of the surrogate model is chosen to be a polynomial fitted to 10 to 15 individual CHIMERE
simulations performed every day for the air quality forecast extending between D+0 and D+2. The choice of a polynomial is for clarity and simplicity, and alternative parametric or non-parametric structures could be explored. The number of training scenarios (10 to 15) is constrained by the operational feasibility to perform multiple chemistry-transport simulations.

Four main activity sectors are desired to be captured by ACT, which correspond in terms of SNAP sectors (Selected
Nomenclature for sources of Air Pollution) to the following:

- AGR: Agriculture (SNAP sector 10: including both crops and livestock)
- IND: Industry (SNAP sectors 1, 3, 4: Combustion in energy and transformation industries, combustion in manufacturing industry, Extraction and distribution of fossil fuels and geothermal energy)
- RH: Residential heating (SNAP sector 2: non-industrial combustion plants)
- TRA: Road transport (SNAP sector 7: urban and non-urban roads and motorways)

At present, the following sectors are therefore excluded from the tool, although they could be included in future versions: SNAP5 (extraction and distribution of fossil fuels and geothermal energy), SNAP6 (Solvent and other product use), SNAP8 (off-road sources and machineries such as railways, shipping and air transport), SNAP9: waste treatment and disposal.

Considering the goal to cover four activity sectors with about 10 to 15 training scenarios, we conclude that the surrogate model will be at most a third order polynomial, less if interaction terms are accounted for.

It should be noted that the structure of the surrogate model ultimately developed is expected to deliver satisfactory performances for all pollutants ($PM_{10}$, $PM_{2.5}$, $O_3avg$, $O_3max$ and $NO_2$). The final objective is to implement the surrogate model in an operational forecasting system with a continuous production, no matter whether the focus
in on ozone or particulate matter episodes. Therefore, the selected structure required to deliver satisfactory performances for a given pollutant (e.g. a higher degree polynomial, including interaction terms), can also yield indirect benefits for the other pollutants.





## 2.4    Emission reduction scenario available for development purposes

For development purposes, we performed an extensive set of CHIMERE simulations over the three air pollution episode selected in 2.1 with various levels of emission reduction (10%, 30%, 60%, 90% and 100%) for each of the four activity sectors (AGR, IND, RH, TRA). In all cases, emissions are applied uniformly over Europe and for all chemical species. In the remainder of the paper, those scenarios will be referred by collating the sector and corresponding emission reduction magnitude, e.g. AGR60 for a 60% reduction of agricultural emissions.

In addition to those emission reduction scenarios for individual sectors, we also explored interactions with scenarios where two sectors are reduced simultaneously. We included emission reductions of 30%/60% and 60%/30% for all pairs as well as 100%/100% reduction. Last, 20%/50% reduction level was also included, so that in total we included 45 emission reduction scenarios in this design phase:

- Reference
- AGR10,TRA10,RH10,IND10
- AGR30,TRA30,RH30,IND30
- AGR60,TRA60,RH60,IND60
- AGR90,TRA90,RH90,IND90
- AGR100,TRA100,RH100,IND100
- TRA20AGR50,TRA30AGR60,TRA60AGR30,TRA100AGR100
- TRA20IND50,TRA30IND60,TRA60IND30,TRA100IND100
- AGR20IND50,AGR30IND60,AGR60IND30,AGR100IND100
- AGR20RH50,AGR30RH60,AGR60RH30,AGR100RH100
- IND20RH50,IND30RH60,IND60RH30,IND100RH100
- TRA20RH50,TRA30RH60,TRA60RH30,TRA100RH100

## 3    Design of the optimal surrogate model

Here we present the various steps towards building the polynomial surrogate model fitted on an optimal set of training scenarios. The complexity of the model increases gradually from a univariate form to a multivariable non-linear polynomial including interactions. But to start with, the response of air quality to a given emission reduction for various locations and episodes are illustrated.





### 3.1 Univariate sensitivity to emission reductions

### 3.1.1 Univariate air quality response at individual location

In this section, we present the sensitivity to emission changes for either $PM_{10}$ or $O_3$ in three target cities (Brussels, Paris and Milano) and for the three types of episodes. We aim to illustrate to what extent the air quality response

is linear for each activity sector and species. At this stage, we only present the response in the complete air quality model (CHIMERE), the ability of a surrogate to reproduce this sensibility will be the focus of the following sections.

In Figure 1, we show the difference, at a given point, between the $PM_{10}$ concentration when reducing emission of a single activity sector and the concentrations in the reference simulation. Those differences can be computed from

various levels of reductions: 0% (reference), 10%, 30%, 60%, 90% and 100%.

The $PM_{10}$ concentration response to incremental emission reductions depends on the date, location, and activity sector. The temporal and spatial sensitivity yield substantial complexity in air quality modelling. But we note here that the relationship to incremental emission changes can be relatively simple, and well fitted by low-order polynomial or even linear relationships.

The residential heating (RH) sector contributes mainly with primary $PM_{10}$ emissions or organic aerosol precursors, and its response is the closest to linearity for both dates and the three selected cities. It is also the case for the traffic (TRA) sector. But for industry (IND), and even more so for agriculture (AGR), there is a clear non-linear response. In most cases, such responses follow a second order polynomial. It is for Milano that the steepness of the quadratic term is largest. And for the first episode (March 2015) which is the episode the most influenced by ammonium

nitrate pollution, the fit is closer to a third order polynomial for Brussel and Paris. Very similar behaviors are found for $PM_{2.5}$, displayed in Supplementary Materials (Figure S.1).

The ozone sensitivity to emission changes is illustrated in Figure 2 for both ozone peaks and ozone daily average. As far as ozone peaks are concerned, a quadratic sensitivity is found for traffic and industrial sources, the main providers of NOx and VOC emissions. For the selected day (which is the peak of the episode), traffic is the main

factor, although industry is sensitive also for Brussels. Note the negative contribution of agriculture which, by providing $NH_3$, sequestrates a fraction of nitrogen oxides to form ammonium nitrate, leading indirectly to decrease ozone levels.





The sensitivity to the traffic sector is very different for ozone daily average because that sector is the main contributor to NOx emissions. The switch from high-NOx to low-NOx regimes can be clearly seen in all three cities in the sensitivity of daily average $O_3$ when the emission reduction exceeds 60% for the traffic sector.

The corresponding figures for $NO_2$ for both a wintertime (December 2016) and summertime (June 2017) episode
are provided in Supplementary Material (Figure S.2) although the response is mostly linear.

### 3.1.2    Univariate model performances over Europe

Here we investigate the selection of the optimal model independently for each activity sector, so that we introduce four univariate models for the activity sectors: AGR, IND, RH, and TRA. The regression is also performed for the four pollutants of interest: $PM_{10}$, $PM_{2.5}$, $O_3max$, $O_3avg$, $NO_2$.

For each day, a polynomial model is fitted at each grid point of the modelling domain. We introduce the following notations for a third order polynomial, with $\alpha_{i,j}, \beta_{i,j}, \gamma_{i,j}$ are the coefficients of the regression (and the later are two nullified for linear or quadratic fits):

$$C_{i,j} - C_{i,j}^{ref} = \alpha_{i,j} \cdot \left(\varepsilon_{i,j} - \varepsilon_{i,j}^{ref}\right) + \beta_{i,j} \cdot \left(\varepsilon_{i,j} - \varepsilon_{i,j}^{ref}\right)^2 + \gamma_{i,j} \cdot \left(\varepsilon_{i,j} - \varepsilon_{i,j}^{ref}\right)^3$$

- $C_{i,j}^{ref}$ is the air pollutant concentration modelled with the CTM for the reference simulation with emissions
$\varepsilon_{i,j}^{ref}$

- $C_{i,j}$ is the air pollutant concentration modelled with the CTM for the reference simulation with emissions $\varepsilon_{i,j}$

- throughout the paper, such regressions will be performed for each *i,j* pair of latitude, longitudes indices in the geographical modelling domain, so that the indices will be dropped in the following notations.

Depending on the model complexity (linear, quadratic or cubic), the model is fitted with one, two or three sensitivity simulations (in addition to the reference). The pairs involving the 0% reduction and any of the other amount of reduction allow performing a linear fit. Triplets involving the 0% reduction and any combination of two other reductions are used to perform a quadratic fit. And an analogous quadruplet is used for cubic fits. Then the model is tested by computing the error of its prediction with respect to the remaining available sensitivity
simulations that were not used in the regression. For the purpose of model development, we performed sensitivity simulations over the three selected case studies with uniform reductions of the emissions for AGR, IND, TRA,



and RH by 10%, 30%, 60%, 90% and 100%. The corresponding list of scenarios available for testing and validation are summarized in Table 1, there are 5, 10, and 10 combinations for the linear, quadratic and cubic forms, respectively.

The error that we discuss here is the absolute difference between the concentration change predicted with the surrogate model for a given emission reduction ($\hat{C} - C^{ref}$) and the corresponding validation Chimere simulation ($C - C^{ref}$) for the same emission reduction ($\varepsilon - \varepsilon^{ref}$). Each fitted model can be tested against several independent Chimere validation simulations (4, 3, and 2, for linear, quadratic and cubic forms, respectively), so that an average of absolute errors is taken ($\hat{C} - C$). We also include a relative error, where a normalization by the corresponding pollutant concentration is used (($\hat{C} - C)/C$).

The errors of the models are computed for each day and grid point, allowing a discussion of the surrogate performances both in terms of spatial and temporal variability.

Figure 3 shows the day-to-day variation of the absolute and relative errors of the various univariate $PM_{10}$ models for Agriculture sector, averaged for all grid points in the region 40N to 55N and 10W to 30E. The time period displayed here spans from 20161130 to 20161229 and includes two important air pollution episodes around 20161208 and 20161220. The models are listed in the same order as in Table 1 with first the 5 linear models, then the 10 quadratic and 10 cubic forms.

The first 5 surrogate models with solid lines and blue colors have a linear structure. They have larger errors than the following more complex models, with relative errors ranging from 1.4 to 4%. The worst linear model is fitted using the 10% emission reduction scenario (and the reference), whereas the linear model fitted with the 90% emission reduction scenario is performing better than several more complex models.

Moving to a second order polynomial improves notably the performances. Ten combinations of training scenarios (and the reference) are available to fit a quadratic polynomial. For each of them, three validation scenarios are available to assess the performances. The best quadratic model for that time period relies on the 60% and 100% emission reductions, its relative error is lower than 0.1μg/m³ or 0.65%. Note that the quadratic models using training scenarios confined within a narrow range of emission reductions (e.g. 10% and 30%, or 90% and 100%) do not perform well, even when compared to the simple linear regressions.



The third order polynomial perform best, except the last model which uses only the largest emission reduction and is therefore too weakly constrained for the lower range of reductions. The smallest error for a cubic model is $0.05\mu g/m^3$ or 0.37%.

At this stage, it becomes clear that a tradeoff must be sought between model performance and complexity
considering that moving from a second to third order polynomial requires computing one more training scenario, whereas the error of the quadratic fit is already below 1%.

The errors presented in Figure 3 are an average over a large fraction of Europe. To check if they do not hide some compensations between different regions, we also show in the map of relative errors, averaged over the whole month of December 2016 (Figure 4). The quadratic model performing best according to Figure 3 is fitted to 60%
and 100% emission reductions. The map of error show that the largest errors are found over the Po-Valley, but it is also the case of all the other quadratic models represented in the Figure. Therefore, we conclude that the selection of the best performing model performed on the basis of average performances in Figure 3 do not include compensation between different regions.

A similar analysis is performed for the other activity sectors, the analogous of Figure 3 for Traffic, Residential
Heating, and Industry are provided in Supplementary Material (Figures S.3 to S.8).

The performances of the univariate model are presented in Figure 5 for $PM_{10}$. The four activity sectors are still treated independently at this stage, the interactions and performances of multivariate models will be discussed in Sections 3.2 and 3.3, respectively. Instead of showing the average error in space or time as in Figure 3 and Figure 4, respectively, we show here the whole distribution of error for any grid point in Western Europe, and any day
over three one-month particulate matter air pollution episodes (201503, 201612 and 201701). The main features of those distributions are given as boxplots with the boxes providing the first quartile, median, and third quartile, and whiskers providing the 95% confidence interval (as 1.58 times the interquartile range divided by the square root of the sample size). The points standing out of the 95% confidence interval are also provided. As in Figure 3 and Table 1, we show (from left to right) first the 5 linear surrogate models, and then the 10 quadratic and 10 cubic
forms. The numerical values of the median of those distributions are given in Supplementary Material (Table S.1).

For each activity sector, we find in general that the linear models do not perform as well as the quadratic or cubic polynomials. But for some activity sectors, the linear model can be considered satisfactory.





It is the case for the Residential Heating sector, where a linear model relying on the scenario with 90% reduction (index 4 in the x-axis of Figure 5) is already very good: the upper 95% confidence interval is lower than 0.1%. The median of relative errors is then 0.03% (Table S.1) and the gain when moving to a quadratic form is only a factor two.

On the contrary, for Industry and Traffic, we opt for a quadratic model (using 60% and 100% reductions, index 14 in the x-axis of Figure 5) which yield median errors below 0.1% (0.099 and 0.028%, for IND and TRA, respectively) and the gain in term of median error is a factor 3-4 compared to the linear forms. For the selected quadratic form and 60% and 100% reduction training scenarios, we can ensure that the relative error of the surrogate model is below 10% for any day and any grid point and even below 0.1% for 75% of the points in the
distribution.

For the impact of the agricultural sector on $PM_{10}$, the non-linearity is such that we have to lower slightly the ambition on the performance of the surrogate. Nevertheless, by selecting a quadratic form trained on the scenario with 60% and 100% reduction (index 14 in the x-axis of Figure 5), we can still ensure that the relative error does not exceed 10% for any day and grid point, and even remains below than 1% for 75% of the points in the
distribution.

The choice to select a quadratic model for Industry and Traffic is further supported by the analysis for ozone (Figure 6, and corresponding numerical values in Supplementary Material Table S.2), where a clear improvement is found compared to linear forms. Selecting a quadratic model trained on 60% and 100% emission reduction scenarios allows reaching relative errors lower than 10% for any day and any grid point, and 95% confidence
interval lower than 0.1%.

A linear model could be fit-for-purpose with regards to ozone sensitivity to agriculture emissions, but since a quadratic form was selected for $PM_{10}$, it will also benefit the ozone models. For Residential Heating, most errors are below 0.001% so that the ozone result only confirms the satisfactory behavior of the linear model.

### 3.2 Bivariate models and interactions

After having introduced quadratic terms, we investigate cross-sector interactions. First, we assess the need to account or not for interaction terms. In the case where the added value of interaction terms is demonstrated, we identify the optimal training scenarios.



The surrogate models that we use here are bi-variate, second order polynomial, plus an interaction term, quadrivariate models will be discussed in 3.3. For instance, for the bivariate model of agriculture and industry, we have the following structure:

$$C^{agr,ind} - C^{ref} = \alpha^{agr} \cdot (\varepsilon^{agr} - \varepsilon^{ref}) + \beta^{agr} \cdot (\varepsilon^{agr} - \varepsilon^{ref})^2 + \alpha^{ind} \cdot (\varepsilon^{ind} - \varepsilon^{ref}) + \beta^{ind} \cdot (\varepsilon^{ind} - \varepsilon^{ref})^2$$
$$+ \gamma \cdot (\varepsilon^{agr} - \varepsilon^{ref}) \cdot (\varepsilon^{ind} - \varepsilon^{ref})$$

In order to assess the need to account for interactions, we only use training scenarios with 30% and 60% reduction levels.

First a bivariate quadratic model without interactions is trained with the 30% and 60% reduction levels and tested against corresponding interaction scenarios. Taking the example of agriculture & industry, we would have the two training and testing configurations in lines 1 and 2 of Table 2.

The boxplots of Figure 7 display the performances when including (first two boxplots of each panel) or excluding (last two boxplots) interaction terms. The numeric values of the median of these distributions are available in Table S.3 and S.4 of the Supplementary Material. As could be expected given the relatively linear behavior of the response to Residential Heating emission changes, interactions do not bring substantial added value for those terms: (AGR,RH), (IND,RH), (TRA,RH). On the contrary, keeping 75% of the points and days with a relative error lower than 1% require to account for interactions for the pairs: (AGR,IND) and (TRA,AGR).

For ozone (Figure 8 and Table S.4 of the Supplementary Material), the only important interaction terms are for the (TRA,IND) pair of sectors, which was expected since those bring the largest share of ozone precursor emissions. When ignoring their interactions, the upper 95[th] confidence interval of relative error distribution can reach 1%, whereas it remains below 0.1% when interactions are taken into account.

For the pairs where interactions must be taken into account: (AGR,IND) (TRA,IND) and (TRA,AGR), we remain to identify the optimal level of reduction in the training scenario. We set the range of reduction identified in Section 3.1.2 for the quadratic terms (60/100%) and seek to identify the optimal reduction for the scenario designed to capture interaction terms. The list of available combinations to train and test each interaction term is given in Table 3.

The 30/60% reductions are optimal for the (AGR,IND) and (TRA,IND) pairs, but for the (TRA,AGR) pair, better performances are found with a 100/100 interaction term. The same feature is found for both $PM_{10}$ and $O_3max$, as





seen in Figure 9 and Figure 10, and corresponding numerical values in Table S.5 and Table S.6 of the Supplementary Material.

### 3.3 Quadrivariate models and interactions

The methodology followed in Sections 3.1 and 3.2 consists in selecting first the optimal structure for univariate models, before investigating bivariate models including interactions terms. Such a step-by-step approach allows a clear introduction of the methodology. However, by doing so, we assume that the optimal structure and training scenario remains valid when including interactions whereas there is a possibility that the addition of an interaction term could change the selection of univariate terms.

Therefore, we investigated directly a 4-dimensional model with the following structure:

$$C^{agr,ind,rh,tra} - C^{ref}$$

$$= \alpha^{agr} \cdot (\varepsilon^{agr} - \varepsilon^{ref}) + \beta^{agr} \cdot (\varepsilon^{agr} - \varepsilon^{ref})^2 + \alpha^{ind} \cdot (\varepsilon^{ind} - \varepsilon^{ref}) + \beta^{ind}$$
$$\cdot (\varepsilon^{ind} - \varepsilon^{ref})^2 + \alpha^{rh} \cdot (\varepsilon^{rh} - \varepsilon^{ref}) + \alpha^{tra} \cdot (\varepsilon^{tra} - \varepsilon^{ref}) + \beta^{tra} \cdot (\varepsilon^{tra} - \varepsilon^{ref})^2 + \gamma^{agr,ind}$$
$$\cdot (\varepsilon^{agr} - \varepsilon^{ref}) \cdot (\varepsilon^{ind} - \varepsilon^{ref}) + \gamma^{tra,agr} \cdot (\varepsilon^{tra} - \varepsilon^{ref}) \cdot (\varepsilon^{agr} - \varepsilon^{ref}) + \gamma^{tra,ind}$$
$$\cdot (\varepsilon^{tra} - \varepsilon^{ref}) \cdot (\varepsilon^{ind} - \varepsilon^{ref})$$

Such a model requires two training scenarios for AGR, IND, TRA, one for RH and one for each of the three interaction terms. We did not investigate all possible of the 46 available testing scenarios but a subset applying the same range of reduction for each of the univariate component (i.e. for instance only AGR30, AGR60, IND30, IND60, RH30, TRA30, TRA60) and complementary reduction for the interaction terms (not using 30/60% reductions for our example). With these constrains, we are left already with an impressive number of 285 combinations. The performance (as median relative error) of the optimal set of training scenarios identified in Sections 3.1 and 3.2 is confirmed here as it ranks respectively 21[st], 4[th] and 7[th] for the PM$_{10}$ episodes of 201503, 201612, 201701. Other combinations of training scenarios can indeed be identified for given episodes, but the choice we propose is also robust across various episodes, and always within 5% of the errors of the optimal model. For ozone daily maximum, the model with the step-by-step methodology beats any of the 285 other configurations.

### 3.4 Final structure and performances

To summarize, the model that we selected is a quadrivariate polynomial, first order for RH emissions, and quadratic for AGR, IND, TRA, with interaction terms for the pairs: (AGR,IND), (TRA,IND) and (TRA,AGR).





The optimal set of training scenarios include the 10 sensitivity simulations selected above, to which a Reference is added as well as a simulation where the emissions of sectors are reduced by 100% to ensure that the model remains bounded:

- Reference
- AGR60, AGR100,
- IND60, IND100
- RH90
- TRA60, TRA100
- AGR30IND60
- TRA30IND60
- TRA100AGR100
- AGR100IND100RH100TRA100

This final model is tested the 34 available scenarios not used in the training. We conclude that with such a model structure and training scenarios, it is possible to reach the surrogate model performances summarized in Figure 11

that shows that relative errors are below 1% at 75% of the grid points and days, below 2% at 95% of the grid points and days, and below 10% for any grid points and days. More specifically, the single highest error over the 864 248 grid points and days considered is 7.5%, 7.9%, 4.8%, 2.8% and 2.9% for the same list of pollutants.

## 4 Scenario analysis in air quality forecasts

### 4.1 Description of ACT interface

The routine production of ACT relies on two main steps. First the 12 training scenarios corresponding to the current air quality forecast are simulated on a high-performance computer with the CHIMERE model with a setup inspired from the French Air Quality Platform Prev'Air (Rouïl et al., 2009) and the European CAMS Regional Production (Marécal et al., 2015). Then the surrogate model is fitted statistically and exported to a web interface using the shiny package of the R language.

An annotated screenshot of the ACT web interface is given in Figure S.9 of the Supplementary Material. First the pollutant of interest is selected in the list of compounds for which the surrogate has been validated: $PM_{10}$, $PM_{2.5}$,





NO₂, O₃max, O₃avg, all of which being daily mean values, except for O₃max which is the daily maximum level. The base time of the forecast can also be changed up with access to a long history as well as its valid time.

A series of slide bars allow the user to define its tailored scenario, reducing by any percentage the emission originating from Road transportation, Industry, Residential heating, and Agriculture. Considering that there are
also other sources of pollutions than those included in ACT (i.e. AGR, IND, RH and TRA), a possibility to include or remove those other contributions is offered in order to visualize a reference simulation including only the sources upon which the user can interact through the slide bars. Such "other contributions" include mainly natural emissions (e.g. dust and sea salt for particulate matter, or biogenic VOCs) but also some activity sectors not included at present in the tool (e.g. international shipping). It is computed by withdrawing from the Chimere
Reference simulation a scenario emulated with ACT with all four activity sectors set to zero. As a consequence, the tropospheric ozone burden is also withdrawn, that is why the corresponding menu refers to "natural and background concentrations".

The results are then displayed as a map for either the selected scenario or the difference compared to the reference simulation.

**4.2        Case studies of Scenario analysis**

The use of the ACT interface is illustrated here taking as example the training episodes introduced in Section 2.1.

### 4.2.1        March 2015

In March 2015, a remarkable PM₁₀ episode spread throughout a large part of Western Europe for almost a week. The daily mean for 20150318 displayed in Figure 12 is given either with all sources included (left), or only for the
sources included in ACT: AGR, IND, RH, TRA. As explained in Section 4.1, the other sources are mainly natural (desert dust and sea salt), but they also include some activity sectors not available in ACT (such as international shipping).

During that episode, the PM₁₀ composition was dominated by inorganic aerosols (Petit et al., 2017), and more specifically ammonium nitrate which is formed by chemical reactions between ammonia and nitrogen oxides
emitted by any combustion sources (traffic or other). In Europe, 93% of annual NH₃ emissions were due to agriculture in 2015 (according to EMEP emissions for EU28 available at www.ceip.at), about half of which were due to livestock and the other half to fertilizer. Fertilizer spreading is a very seasonal activity, that dominates in March. Because NH₃ resulting from fertilizer spreading is emitted over very large areas, and because of the lifetime





of fine PM in the atmosphere, the resulting air pollution plume can reach a substantial geographical extent as we can see here with high $PM_{10}$ levels modelled far out over the Atlantic Ocean. This type of air pollution event is a textbook example of the regional character of atmospheric air pollution.

In order to illustrate the capacities of the ACT tool, four scenarios are emulated by removing independently 100% of the emission of each of the four main activity sectors (Figure 12). These scenarios are all emulated on the basis of the simulation excluding natural sources, i.e. to be compare to the top right panel of Figure 12.

The reduction of agricultural emissions has the largest effect on $PM_{10}$ concentrations and only a couple of hotspots remain, for instance in the Pô Valley. On the contrary, none of the scenario where only one of the other sectors is reduced manages to reach low $PM_{10}$ levels. This is because $NH_3$ remains in excess and removing all the NOx from Traffic has little effect if the NOx from residential heating remains available. We can note however that the punctual sources in Turkey or Ukraine disappear in the scenario where Industrial emissions are set to zero. Last, industrial emissions have a larger effect on regional pollution than traffic, which seems contradictory to the high attention given to road transportation during major air pollution episodes.

Last, we also display an emulated scenario where both traffic and industrial emissions are set to zero, whereas agriculture emission are unchanged. Here we obtain low $PM_{10}$ levels similar to the no-agriculture scenario because most sulfur and nitrogen oxides are removed.

### 4.2.2    December 2016

In December 2016, a large particulate matter episode developed in a Western Europe under the influence of cold and stable meteorological conditions that kept air pollutants near the ground in the inversion layer (Forêt et al., 2017). In addition, cold temperatures induced an increase in residential heating emission. The reference simulation for daily mean $PM_{10}$ on 20161201 is presented in Figure 13. It only includes the sources of the main activity sectors (e.g. excluding natural and background concentrations).

High $PM_{10}$ concentrations are modelled over Northern Italy, but also a large part of Northern France as well as south-western UK. There are also some scattered areas of Pollution in Eastern Ukraine. The removal of emission from each of the four main activity sectors is emulated with ACT in Figure 13, using as reference the simulation excluding natural sources. All sectors have an influence on $PM_{10}$ concentrations, but the largest contribution is attributed to agriculture, which is really the only source that has an impact on background $PM_{10}$, although some hotspots remain. According to the scenario where a 100% reduction of industrial emission in emulated, it seems





that the $PM_{10}$ peak in Eastern Ukraine is due to industrial activities. The hotspot in Paris and Milano remain to some extent in all for the four scenarios, demonstrating that air pollution in those areas can only be mitigated by acting on all activity sectors.

The large role of agriculture is due to the high sensitivity of atmospheric chemistry to ammonia ($NH_3$) emissions which reacts with nitrogen oxides emitted from any combustion source (traffic or other) to form ammonium nitrate. Given the strong seasonality of fertilizer spreading, ammonia emissions December is likely due mainly to livestock emissions. The ACT results illustrate clearly the importance of Agriculture for PM air pollution by allowing to emulate a scenario by removing all $NH_3$ emissions. One should however keep in mind the challenge in mitigating $NH_3$, where the emission reduction between 2005 and 2020 is only 1 to 24% in the Gothenburg protocol depending on the country (Bessagnet et al., 2014).

### 4.2.3    June 2017

For ozone, we selected the intense episode of June 2017 (Figure 14), (Tarrason et al., 2017). An ozone anomaly above 60 to 70µg/m$^3$ was due to European emissions of the four main activity sectors, but Figure 15 also highlights the importance of tropospheric burden for ozone air pollution where a large fraction cannot be mitigated by reducing European emissions alone.

The analysis of emission reduction responses (Figure 14) is much more predictable than the impact on PM. Reducing emissions from residential and agriculture sectors has almost no impact on ozone concentrations. Ozone is driven by emissions from traffic (nitrogen oxides) and from industry (nitrogen oxides and volatile organic compounds). For the case studied, removing traffic emissions decreases ozone concentrations by 20 to 60 µg/m$^3$ in most of the places, and the impact is not as strong with a 100% reduction in industry emissions.

### 5    Source allocation mode

The surrogate ACT model can also be used in source allocation mode. By reducing successively each activity sector by 100%, it is possible to compute its contribution to the burden of air pollution for a given day and location. Here the contribution would be the gain in concentration reduction induced by removing emissions from a given activity sector. It may differ substantially from a source apportionment approach where the objective is to assess the contribution - in mass – of a sector to the overall PM burden. A good illustration of such differences, is the underestimation of the influence of a sector such as agriculture that only contributes with relatively light



compounds in terms of molecular weight ($NH_3$) but are very sensitive in the formation of secondary PM. The methodological difference and purposes are explained for instance in (Clappier et al., 2017a) that also emphasize the role of interaction terms further illustrated below.

Unlike source apportionment, the allocation we display here indeed shows the reduction of concentration that can be achieved by removing totally the emissions of an activity sectors. Such results are generally obtained by zeroing out anthropogenic emissions in a full CTM simulation. It can also be extrapolated from sensitivity simulation based only on 15 to 30% emission reductions. But then the uncertainties become substantial in the case of non-linear response. The structure of ACT, by being fitted and tested for emission reductions ranging from 0 to 100% offers a more reliable response in that context.

For the two $PM_{10}$ episodes and the $O_3$ episode introduced in Section 2.1 and for the city of Paris, we isolate in Figure 15 the impact of each of the four activity sectors as well as natural and background concentrations. A specific focus is also given on how interaction terms are handled in this decomposition. On the left column, the gap between individual sectors and the overall reduction is explicitly provided as an interaction term. Here, the contribution of each activity sector is assess by setting its emissions to zero (referred to as top-down brute force method in (Clappier et al., 2017a)). Interactions are computed by difference between the sum of individual contribution and a scenario emulated with all four activity sector removed simultaneously (which is constrained by the scenario "AGR100IND100RH100TRA100", Section 3.4). And the remaining fraction corresponds to natural sources or any other pollutant precursor not included in ACT (natural and background concentration).

From this decomposition, the large influence of Agriculture appears clearly for the March 2015 $PM_{10}$ episode in Paris (Figure 15). Traffic and Industry are also important contributors, but residential heating has a smaller contribution. For the December 2016 $PM_{10}$ episode, the agriculture contribution is only second behind traffic. Residential heating is more important than for the March 2015 episode, but it could also be underestimated by the model, which used emission inventory that do not capture very well residential heating, in particular in relation with wood burning.

For ozone, the picture is very different, a large fraction of ozone is actually attributed to the tropospheric burden. But during the ozone air pollution event (14[th] and 19-22 June), a larger contribution of traffic is found, whereas the impact of industry is very small (a different conclusion will hold for other cities as presented in Section 6). This presentation for ozone highlights very well the challenge of mitigating background levels on the basis of





European emission mitigation. Conversely, it also shows precisely the need to act on European emission during the main ozone peak.

When interactions are negative, the sum of individual contribution of each activity sector exceed the total ("Net") in the reference simulation. It is clearly the case for the strong $PM_{10}$ episode of March 2015, but conversely the interactions can also be positive as illustrated for the $O_3$ episode of June 2017. The importance of interactions was expected considering the complexity of atmospheric chemistry. But it constitutes an artefact that must be dealt with when performing a source allocation by treating each sector independently.

There is no fully satisfactory approach to handle interaction terms in such a decomposition. The simplest alternative is to distribute one-fourth of those interactions into each of the four contributions which leads to re-scaling the reference simulation (Table 4). The contribution of individual sector can change substantially, which is also a reminder for the overall uncertainty of the approach. For instance, the share of Agriculture is reduced from 32% to 23% in the case of the 201503 PM episode. But at least the ranking of each sector is not changed and their qualitative evolution displayed in the right column of Figure 15 is similar.

## 6  Chemical regimes

The surrogate ACT model trained on CHIMERE sensitivity simulations also allows exploring the chemical sensitivity (or regimes) within the parameter space of sectoral emission reductions. ACT is a quadrivariate second order polynomial with interactions using as predictors the four sectors considered. By plotting the surface response to two of these four sectors in a 2D parameter space, it is possible to assess chemical regimes for a given day, location and pollutant. In doing so, we perform an analogy with the classical ozone production isopleths of (Sillman, 1999), by substituting the NOx and VOC emissions in the x and y axes by different activity sectors.

Figure 16 compares two particulate pollution days, over Paris areas, in March 2015 and December 2016, respectively. In March 2015, reducing agriculture emissions has a strong positive impact on the reduction of $PM_{10}$ concentrations that decrease sharply while the impact of traffic emissions reduction is much less effective with isopleth close to vertical lines. The inverse conclusions can be drawn for the December 2016 episode, where a linear decrease of $PM_{10}$ is induced by traffic emission reductions, whereas the isopleths are closer to horizontal lines, depicting a low sensitivity to agricultural emission changes.





Ozone chemical regimes can also be investigated for a high ozone episode (20170621) , for various locations and for both ozone daily maxima and daily means (Figure 17).

Chemical regimes leading to the formation of high ozone values (as the slope of isopleths for daily maximum ozone) are quite similar over Paris and Milano. But in Brussel a stronger sensitivity to Industrial emissions is found. The isopleths for daily average ozone on the same day are very different. For all levels of industrial emissions, we find that a decrease of traffic emissions leads first to an increase of ozone before becoming efficient for the largest levels of emission reduction.

## 7        Conclusion

We presented the first surrogate air quality model designed to explore custom air pollution mitigation scenarios in the every-day air quality forecast. This tool applies for $PM_{10}$, $PM_{2.5}$, $O_3$ (both as daily mean and daily maximum) and $NO_2$ and covers the following activity sectors: agriculture, industry, road transportation and residential heating. It can be implemented within an operational air quality forecasting system and operated interactively by any user through a web interface[2].

Because of the complexity of atmospheric chemistry and physics, chemistry-transport models are required to account for the fate of air pollutants in the atmosphere. Simplified models have been developed over the past for assessment purposes, for instance to identify optimal long-term mitigation strategies. However, such simplified models rely on assumptions which are not valid over short time periods, such as the linearity of the response of air concentrations to incremental emission changes.

We introduce a new surrogate modelling approach, whose main strength is to apply for short time scales, so that it can be embedded in an air quality forecast system. This challenge is achieved by fitting every day a new surrogate model on the basis of the forecast of the corresponding day. Most of the complexity of atmospheric processes remain therefore represented within the full chemistry-transport model, and the only purpose of the surrogate is to offer flexibility.

First, we investigated the non-linearity of the response of atmospheric pollutant concentrations to incremental emission changes for various pollutants, areas and different episode typologies. We concluded that whereas the

---

[2] https://policy.atmosphere.copernicus.eu/CAMS_ACT.php

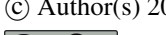



response was mainly linear for residential heating, non-linearities were important, especially for agriculture emissions and their impact on $PM_{10}$ formation, and traffic and industrial emissions for ozone pollution.

The numerical experiment plan to identify the best model structure and the corresponding optimal set of training scenarios is presented by increasing level of complexity. We ultimately select a quadrivariate polynomial of first order for residential heating emissions, and second order for agriculture, industry and traffic emissions with three interaction terms. The surrogate is trained on 10 sensitivity simulations, to which a reference and a closure simulation must be added. With such a structure, we can ensure that relative errors remain below 2% at 95% of the grid points and days for $PM_{10}$, $PM_{2.5}$, $NO_2$, $O_3$max and $O_3$avg.

The user interface is available online and a few case studies are presented. The emulation of custom scenarios is introduced for two $PM_{10}$ and one ozone episode. It highlights the important role of agricultural emission in the formation of regional scale PM episodes, although several hotspots can only be mitigated by acting on all sectors. For ozone, the tropospheric burden is important but during a strong air pollution episode, action on European sources of traffic and industry can reduce peak levels.

The surrogate model can also be used for source allocation, although it requires additional assumptions on the way interaction terms are handled. We also present an innovative application for chemical regime analysis for both ozone and particulate air pollution provides new insight in the identification of the most efficient activity sector to be targeted for air pollution episode mitigation.

At present the main limitation of ACT is that it relies on emission reductions uniform over Europe. Adding geographical flexibility is one of the priorities for further development.

To our knowledge, this model is the first surrogate, or emulator, able to cover short time scale for air pollution studies. Although the structure of the model is determined once by the outcome of the present study, the surrogate is fitted every day to a new air quality forecast therefore paving the way to further develop machine learning in the field of air quality forecasting.

## 8    Code Availability

The script used for the operational daily training of the ACT surrogate is available at https://github.com/acolette/ACT_v1.0. The underlying CHIMERE Chemistry Transport Model is available at https://www.lmd.polytechnique.fr/chimere/chimere.php.



## 9     Author Contributions

AC conceptualised the model, designed the experiment and performed the simulations with support of FM. VL and BR designed the interface of the web toolbox. AC and LR prepared the manuscript with contributions from all co-authors.

## 10     Acknowledgements:

The present work was funded under the Copernicus Atmosphere Monitoring Service Policy Support contract (CAMS_71), also benefiting from the support of the French Ministry in Charge of Environment. The high performance simulations were performed on the Centre de Calcul Recherche et Technologie.

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





**Figure 1 : Modelled PM₁₀ reduction (y-axis : positive for a decrease with respect to the reference, μg/m³) for a given reduction in Agriculture (green), Industrial (red), Residential Heating (blue), and Traffic (black) emissions (x-axis: in %) in Brussels, Paris and Milano (top to bottom) and for 20150320 (left) and 20161201 (right).**



**Figure 2 : Same as Figure 1 for ozone daily maximum (O₃max) and daily average (O₃avg) on 20170620.**



| Model | | training | testing |
|---|---|---|---|
| Linear | 1 | Ref, 10% | 30%, 60%, 90%, 100% |
| | 2 | Ref, 30% | 10%, 60%, 90%, 100% |
| | 3 | Ref, 60% | 10%, 30%, 90%, 100% |
| | 4 | Ref, 90% | 10%, 30%, 60%, 100% |
| | 5 | Ref, 100% | 10%, 30%, 60%, 90% |
| Quadratic | 6 | Ref, 10%, 30% | 60%, 90%, 100% |
| | 7 | Ref, 10%, 60% | 30%, 90%, 100% |
| | 8 | Ref, 10%, 90% | 30%, 60%, 100% |
| | 9 | Ref, 10%, 100% | 30%, 60%, 90% |
| | 10 | Ref, 30%, 60% | 10%, 90%, 100% |
| | 11 | Ref, 30%, 90% | 10%, 60%, 100% |
| | 12 | Ref, 30%, 100% | 10%, 60%, 90% |
| | 13 | Ref, 60%, 90% | 10%, 30%, 100% |
| | 14 | Ref, 60%, 100% | 10%, 30%, 90% |
| | 15 | Ref, 90%, 100% | 10%, 30%, 60% |
| Cubic | 16 | Ref, 10%, 30%, 60% | 90%, 100% |
| | 17 | Ref, 10%, 30%, 90% | 60%, 100% |
| | 18 | Ref, 10%, 30%, 100% | 60%, 90% |
| | 19 | Ref, 10%, 60%, 90% | 30%, 100% |
| | 20 | Ref, 10%, 60%, 100% | 30%, 90% |
| | 21 | Ref, 10%, 90%, 100% | 30%, 60% |
| | 22 | Ref, 30%, 60%, 90% | 10%, 100% |
| | 23 | Ref, 30%, 60%, 100% | 10%, 90% |
| | 24 | Ref, 30%, 90%, 100% | 10%, 60% |
| | 25 | Ref, 60%, 90%, 100% | 10%, 30% |

**Table 1 : list of CTM sensitivity simulations used to train and test the various linear, quadratic and cubic forms of the surrogate model**



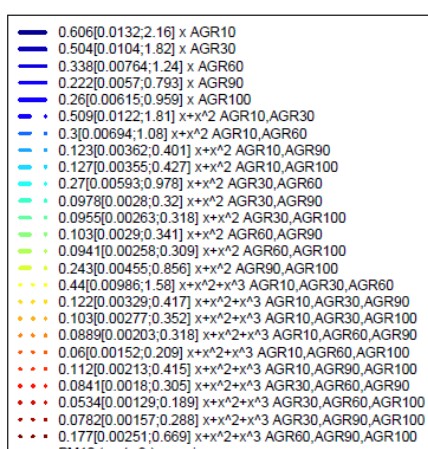

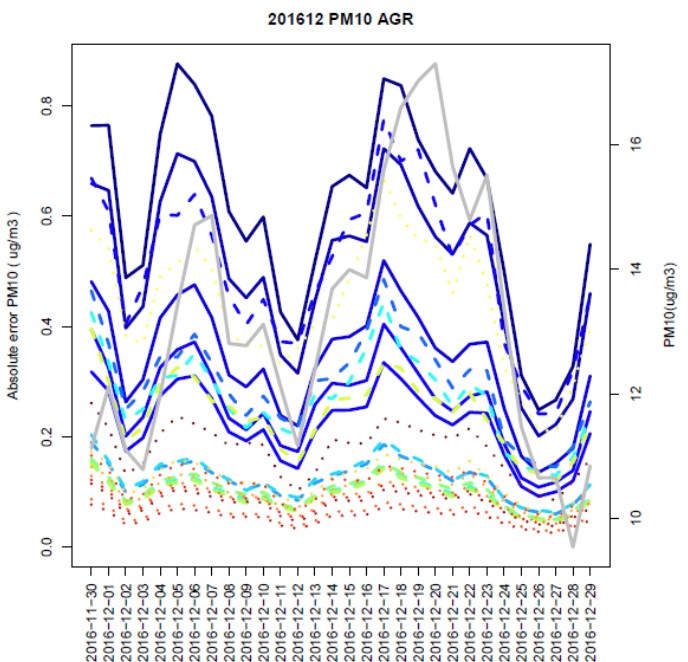



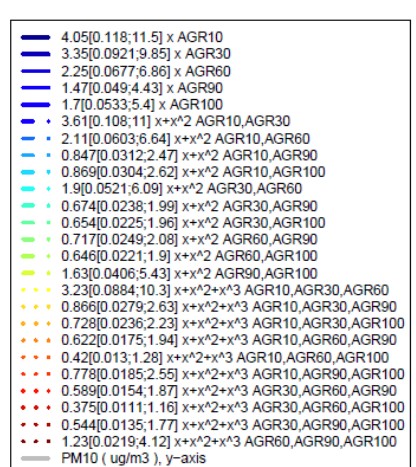

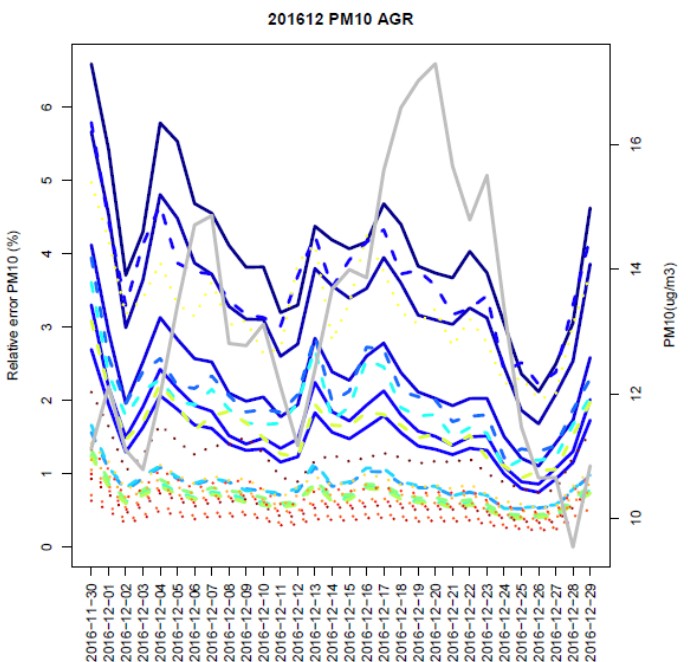

**Figure 3 : Absolute (top, μg/m$^3$) and relative (bottom, %) error over Western Europe of the univariate surrogate model for the Agriculture activity sector in December 2016. The coloured lines are for individual surrogate models, with the complexity and training scenario provided in the legend, as well as the error averaged over the whole time-period. The grey curve gives the day-to-day variation of PM$_{10}$ (μg/m$^3$) averaged over the region (displayed on the right-hand-side y-axis).**





**Figure 4 : Relative error (%) averaged over the month of December 2016 for the quadratic univariate PM$_{10}$ models with respect to the Agriculture activity sector. The sensitivity scenarios used to train the individual models are indicated in the title of each panel, as well as the average error.**







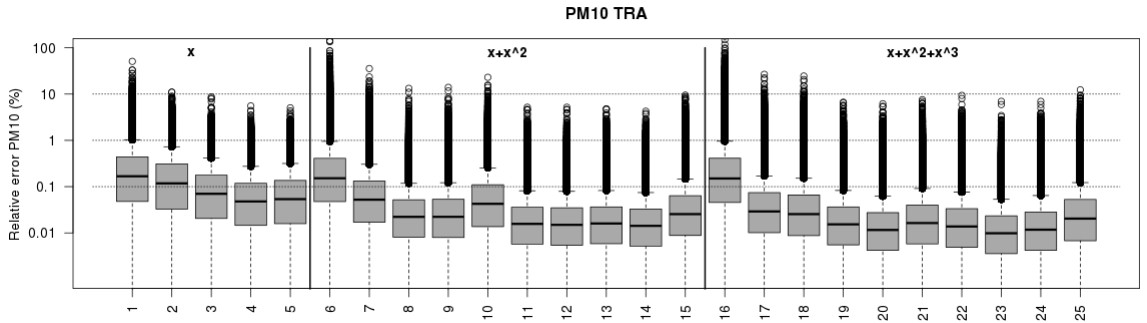

**Figure 5 :** Relative error (%) of the PM$_{10}$ univariate surrogate models for either AGR, IND, RH, TRA (from top to bottom) and for various polynomial forms and training scenarios (x-axis: 5 linear, 10 quadratic and 10 cubic forms with indices of the x-axis matching the rows of Table 1). The boxplots indicate the minimum, first quartile, median, third quartile, and maximum in the distribution of relative errors at each grid points in Western Europe and each day over three air pollution events (201503, 201612, 201701). The dotted horizontal lines are for 0.1%, 1%, and 10% errors.

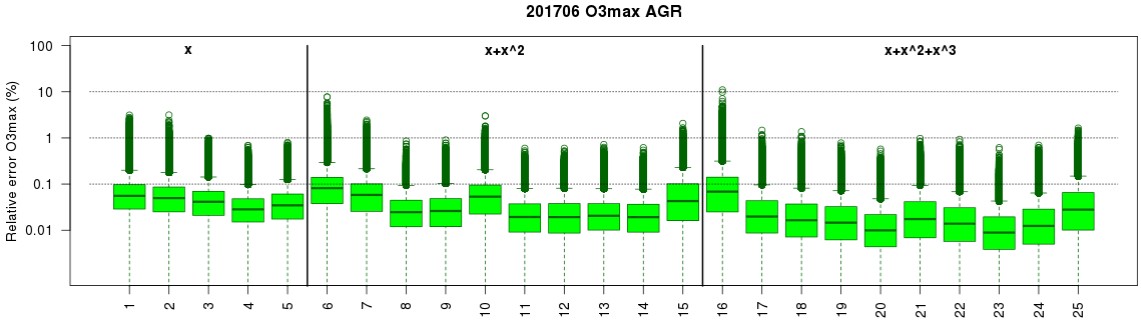

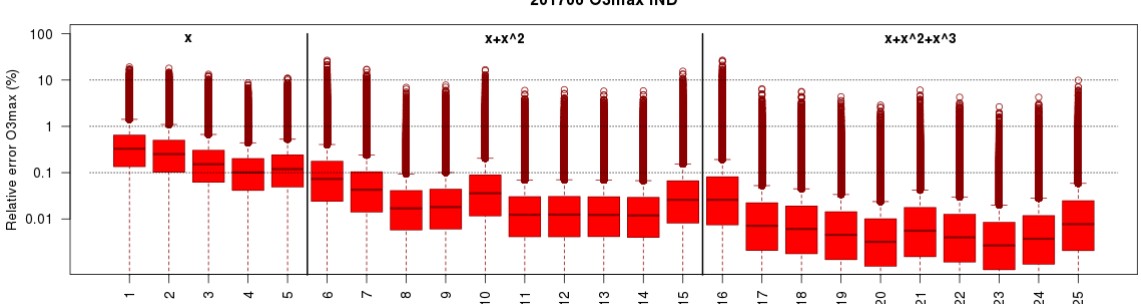



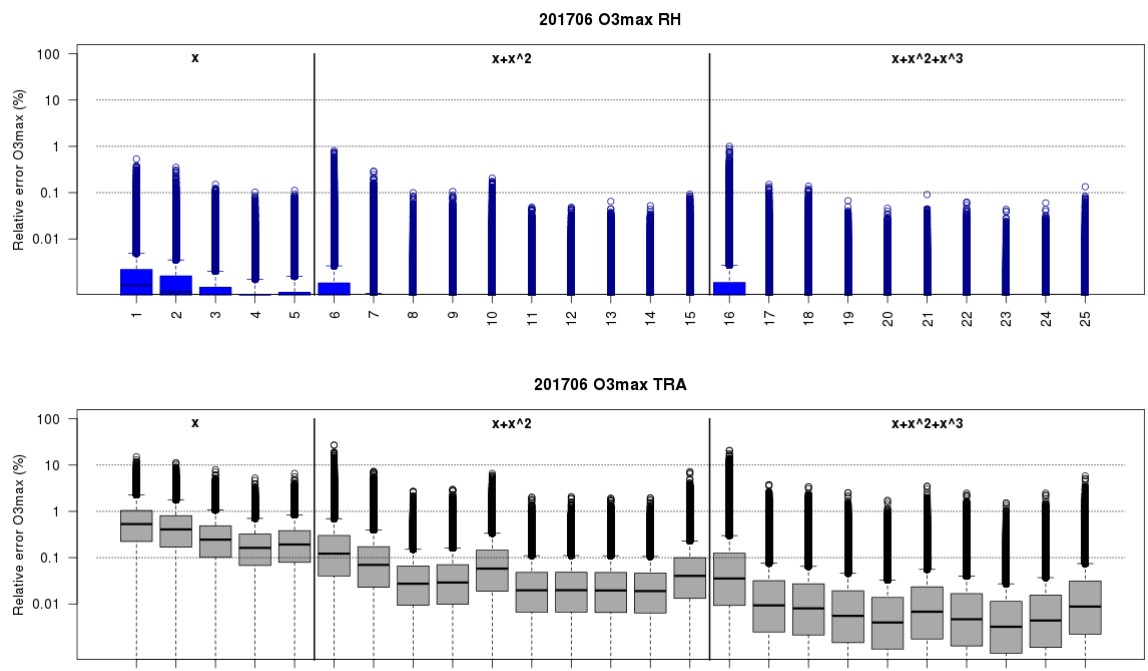

**Figure 6 : Same as Figure 5 for O₃max during the 201706 episode.**

| Model | | training | testing |
|---|---|---|---|
| Interaction | 1 | Ref, AGR30, AGR60, IND30 IND60, AGR60IND30 | AGR30IND60 |
| | 2 | Ref, AGR30, AGR60, IND30 IND60, AGR30IND60 | AGR60IND30 |
| No Interaction | 3 | Ref, AGR30, AGR60, IND30 IND60 | AGR30IND60 |
| | 4 | Ref, AGR30, AGR60, IND30 IND60 | AGR60IND30 |

**Table 2 : list of CTM sensitivity simulations used to train and test the need to account for interactions in the quadratic forms of the surrogate model.**

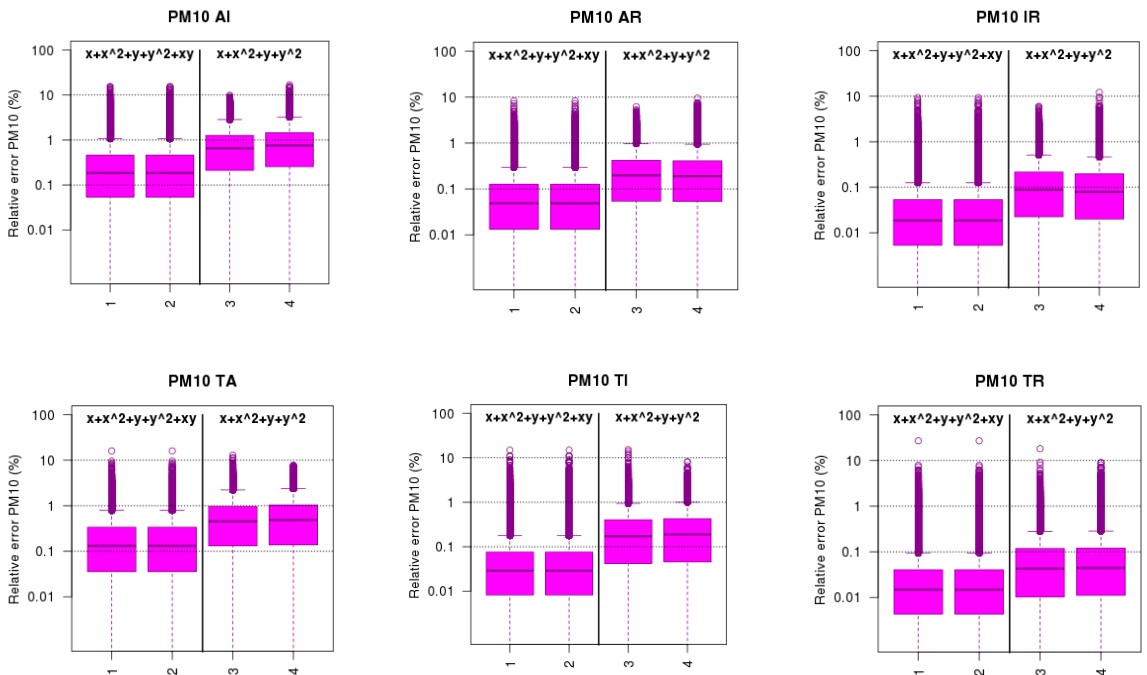

**Figure 7: Relative error (%) of the PM₁₀ bivariate surrogate models (from left to right and top to bottom: AI: AGR&IND, AR: AGR&RH, IR: IND&RH, TA: TRA&AGR, TI: TRA&IND, TR: TRA&RH). For each panel, the boxplots of the distribution of errors are given for 2 models with interactions (left) and 2 models without interactions (right), where indices in the x-axis match the raw of Table 2. The distributions of relative errors include each grid points in Western Europe and each day over three air pollution events (201503, 201612, 201701). The dotted horizontal lines are for 0.1%, 1%, and 10% errors.**

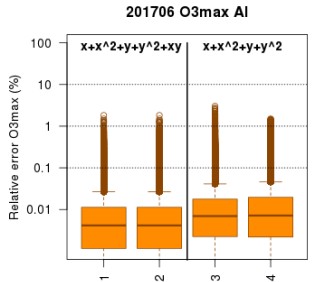 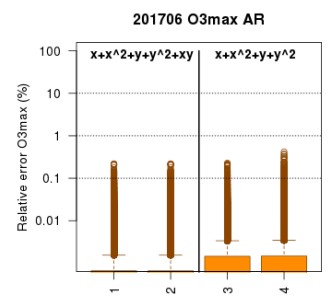 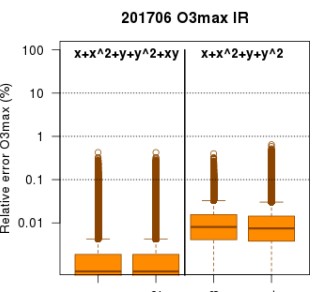





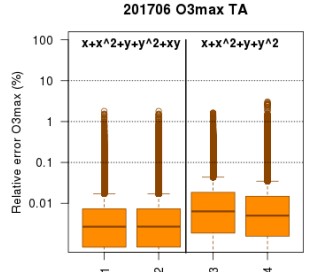 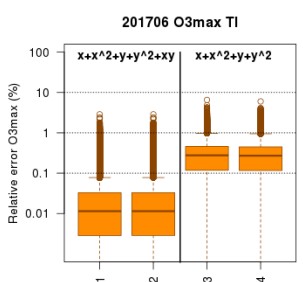 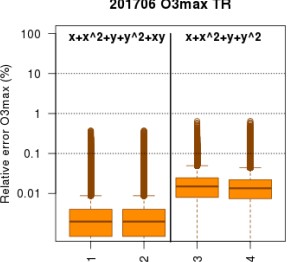

**Figure 8: Same as Figure 7 for O₃max and 201706.**





| Model | | training | testing |
|---|---|---|---|
| Interaction | 1 | Ref, AGR60, AGR100, IND60, IND100, AGR20IND50 | AGR30IND60, AGR60IND30, AGR100IND100 |
| | 2 | Ref, AGR60, AGR100, IND60, IND100, AGR30IND60 | AGR20IND50, AGR60IND30, AGR100IND100 |
| | 3 | Ref, AGR60, AGR100, IND60, IND100, AGR60IND30 | AGR20IND50, AGR30IND60, AGR100IND100 |
| | 4 | Ref, AGR60, AGR100, IND60, IND100, AGR100IND100 | AGR20IND50, AGR30IND60, AGR60IND30 |

**Table 3 : list of CTM sensitivity simulations used to select the optimal scenario by training and testing the various combinations to account for interactions in the quadratic forms of the surrogate model.**

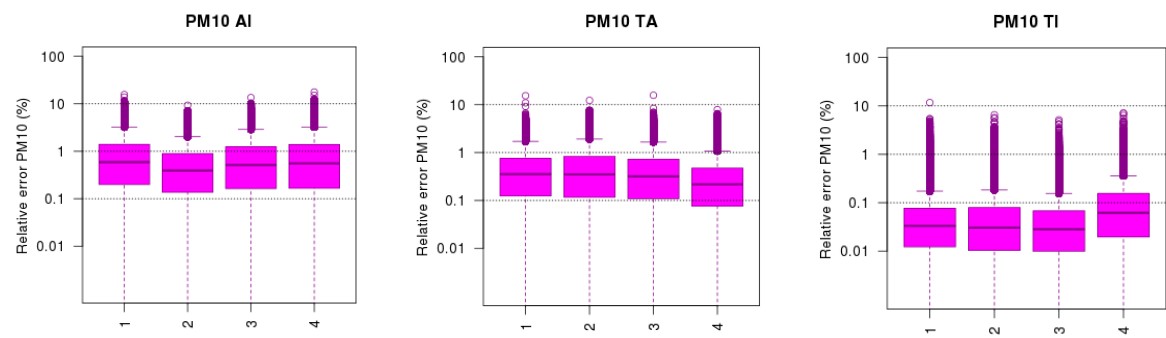

**Figure 9: Relative error (%) of the $PM_{10}$ bivariate surrogate models (from left to right and top to bottom: AI: AGR&IND, TA: TRA&AGR, TI: TRA&IND). For each panel, the boxplots of the distribution of errors are given for 4 models with interactions trained on different scenarios where indices in the x-axis match the raw of Table 3Table 2. The distributions of relative errors include each grid points in Western Europe and each day over three air pollution events (201503, 201612, 201701). The dotted horizontal lines are for 0.1%, 1%, and 10% errors.**





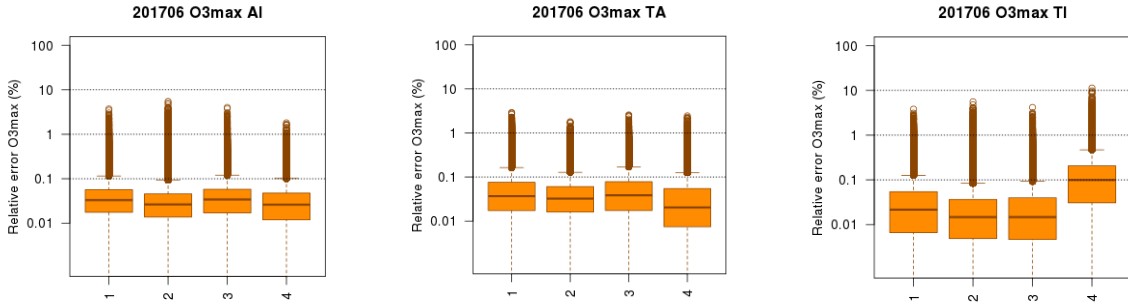

**Figure 10: Same as Figure 9 for O₃max and 201706.**

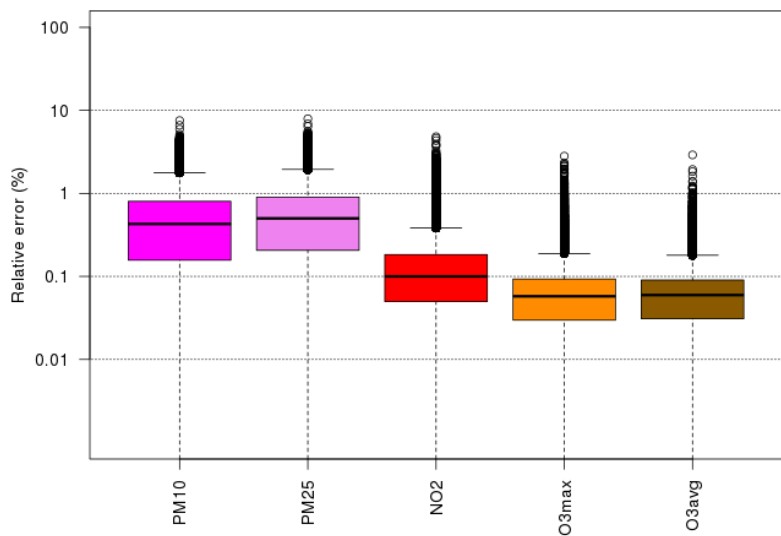

**Figure 11: Relative error of the final selected surrogate model for all pollutants. The boxplots represent the distribution of relative errors over each grid points in Western Europe and each day during the relevant air pollution events: wintertime (2015903, 201612, 201701) for PM₁₀, PM₂.₅ and NO₂, summertime (201706) for O₃max (i.e. daily maximum ozone) and O₃avg (daily average).**





PM10 - daily mean ($\mu g/m^3$)

Reference PM10 map including all sources (natural and anthropogenic) and background concentrations.

PM10 - daily mean ($\mu g/m^3$)

PM10 map including only the main anthropogenic sources: (agriculture, industry, trafic and residential heating). (mainly natural dust and sea salt for particulate matter) but also a few anthropogenic sectors such as shipping.

PM10 - daily mean ($\mu g/m^3$)

Map of PM10 for a Europe-wide uniform reduction of:
Agriculture: 0 % ; Traffic: 100 % ; Residential: 0 % ; Industry: 0 %

PM10 - daily mean ($\mu g/m^3$)

Map of PM10 for a Europe-wide uniform reduction of:
Agriculture: 0 % ; Traffic: 0 % ; Residential: 100 % ; Industry: 0 %

PM10 - daily mean ($\mu g/m^3$)

Map of PM10 for a Europe-wide uniform reduction of:
Agriculture: 0 % ; Traffic: 0 % ; Residential: 0 % ; Industry: 100 %

PM10 - daily mean ($\mu g/m^3$)

Map of PM10 for a Europe-wide uniform reduction of:
Agriculture: 100 % ; Traffic: 0 % ; Residential: 0 % ; Industry: 0 %



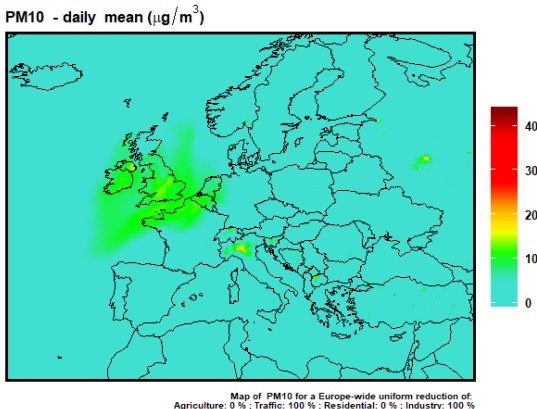

**Figure 12: First row: PM$_{10}$ concentrations on 20150318 including (left) or excluding (right) natural and background concentrations. Following rows: PM$_{10}$ daily average concentrations emulated with ACT for 20150318 for a 100% reduction of traffic emissions (2$^{nd}$ row, left), residential emissions (2$^{nd}$ row, right), industry emissions (3$^{rd}$ row, left), agriculture emissions (3$^{rd}$ row, right), and both traffic and industry (4thr row, right).**

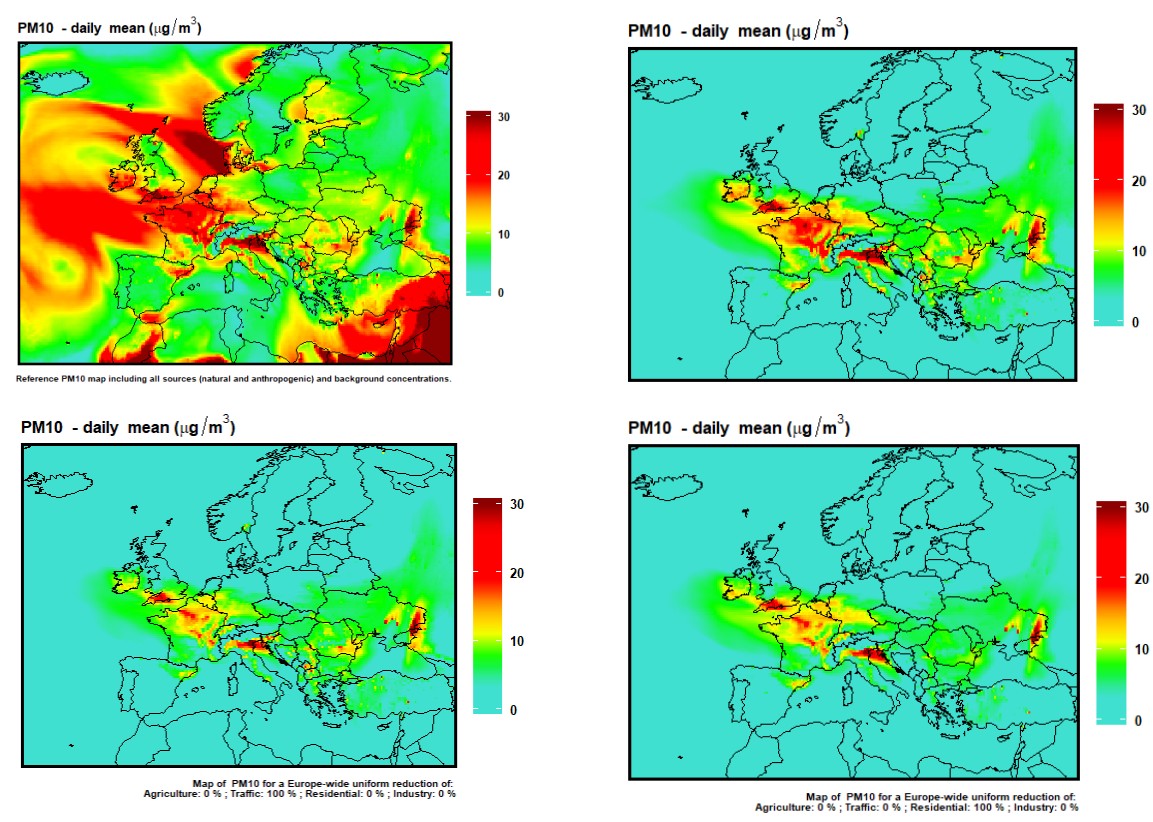

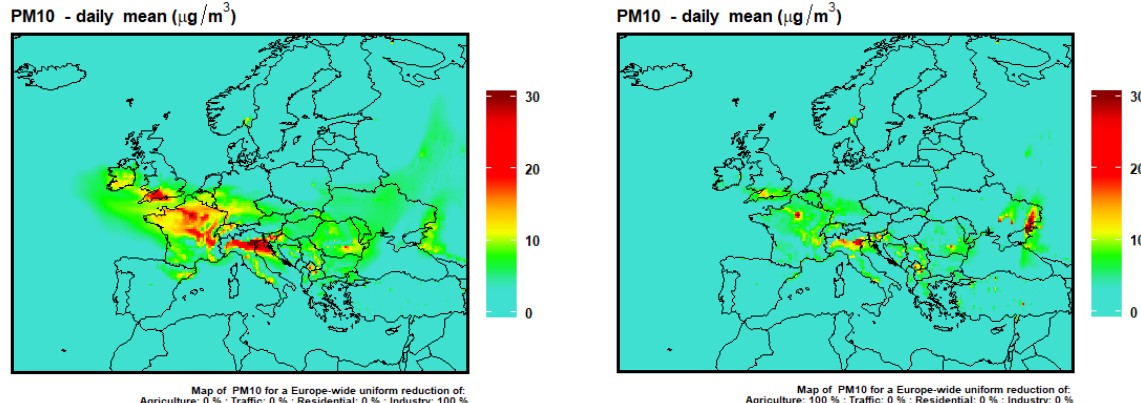

**Figure 13: First row: PM$_{10}$ concentrations on 20161201 including (left) or excluding (right) natural and background concentrations. Following rows: PM$_{10}$ daily average concentrations emulated with ACT for 20161201 for a 100% reduction of traffic emissions (2$^{nd}$ row, left), residential emissions (2$^{nd}$ row, right), industry emissions (3$^{rd}$ row, left), agriculture emissions (3$^{rd}$ row, right).**





**Figure 14: First row: O₃max concentrations on 20170621 including (left) or excluding (right) natural and background concentrations. Following rows: O₃max concentrations emulated with ACT for 20170621 for a 100% reduction of traffic**



emissions (2nd row, left), residential emissions (2nd row, right), industry emissions (3rd row, left), agriculture emissions (3rd row, right).





|  | PM$_{10}$, 201503 | | PM$_{10}$, 201612 | | O$_3$max, 201706 | |
|---|---|---|---|---|---|---|
|  | Explicit interaction | Redistributed interactions | Explicit interaction | Redistributed interactions | Explicit interaction | Redistributed interactions |
| AGR | 32 | 23 | 24 | 21 | -1 | 0 |
| IND | 17 | 11 | 11 | 10 | 1 | 4 |
| RH | 9 | 15 | 16 | 17 | 0 | 1 |
| TRA | 18 | 23 | 24 | 23 | 7 | 10 |
| Other | 24 | 28 | 24 | 28 | 92 | 85 |

**Table 4: Relative contributions (%) of the four main activity sectors as well as natural and background concentrations ("other") averaged over the three selected air pollution episodes, either with explicit interaction terms, or with interactions redistributed within individual contributions.**

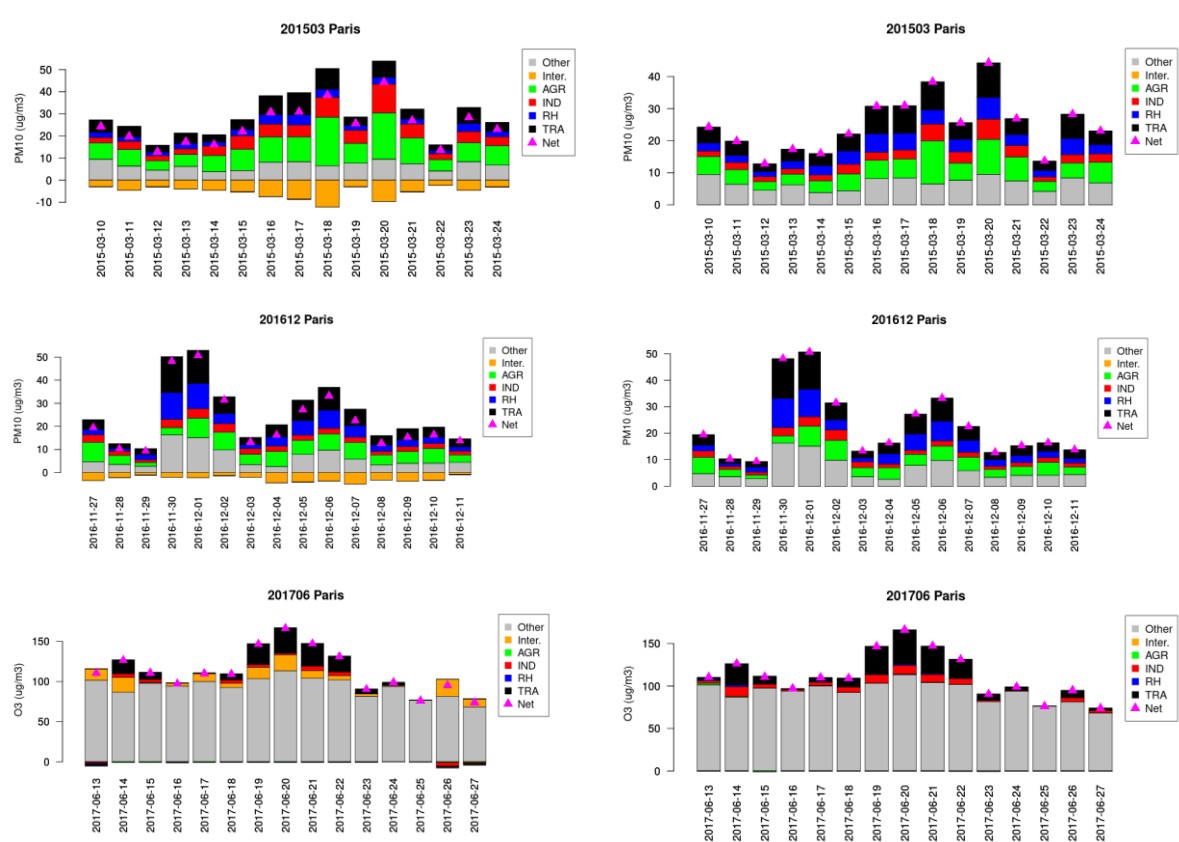



**Figure 15 : Source allocation using the ACT surrogate model for PM$_{10}$ in 201503 (top) and 201612 (middle) and for O$_3$max in 201706 as in absolute levels (µg/m$^3$) with explicit (left) or redistributed (right) interaction terms.**

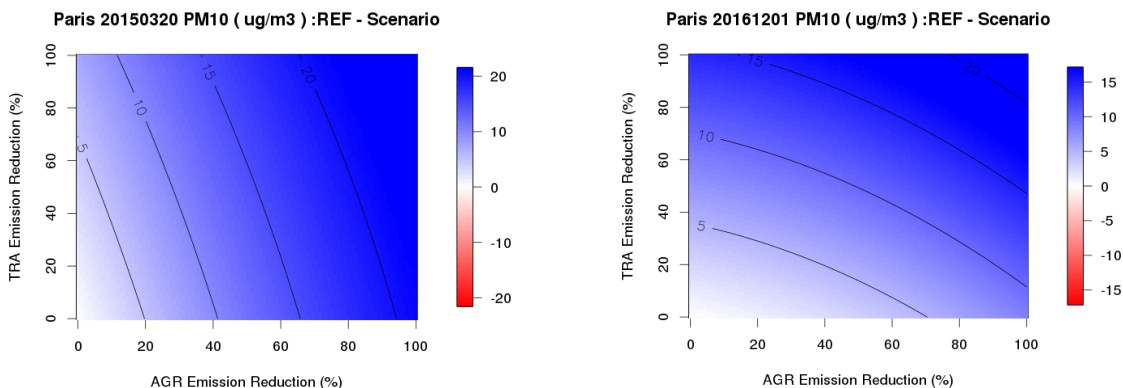

**Figure 16: PM$_{10}$ concentration reduction (positive for a decrease in blue) corresponding to a given reduction in Traffic and Agriculture emissions over Paris area – daily average for 20th March 2015 (left) and 1st December 2016 (right)**

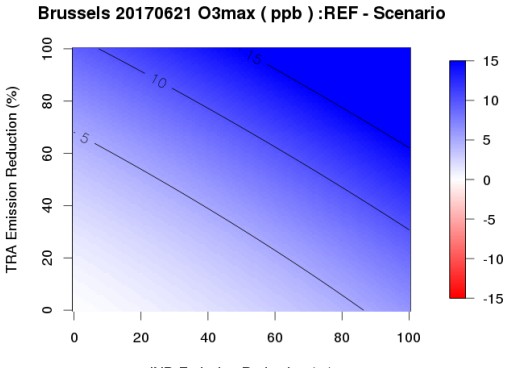
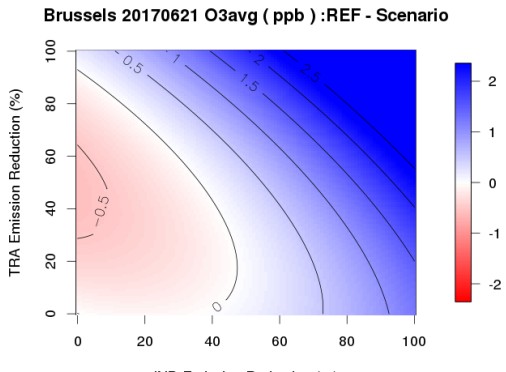



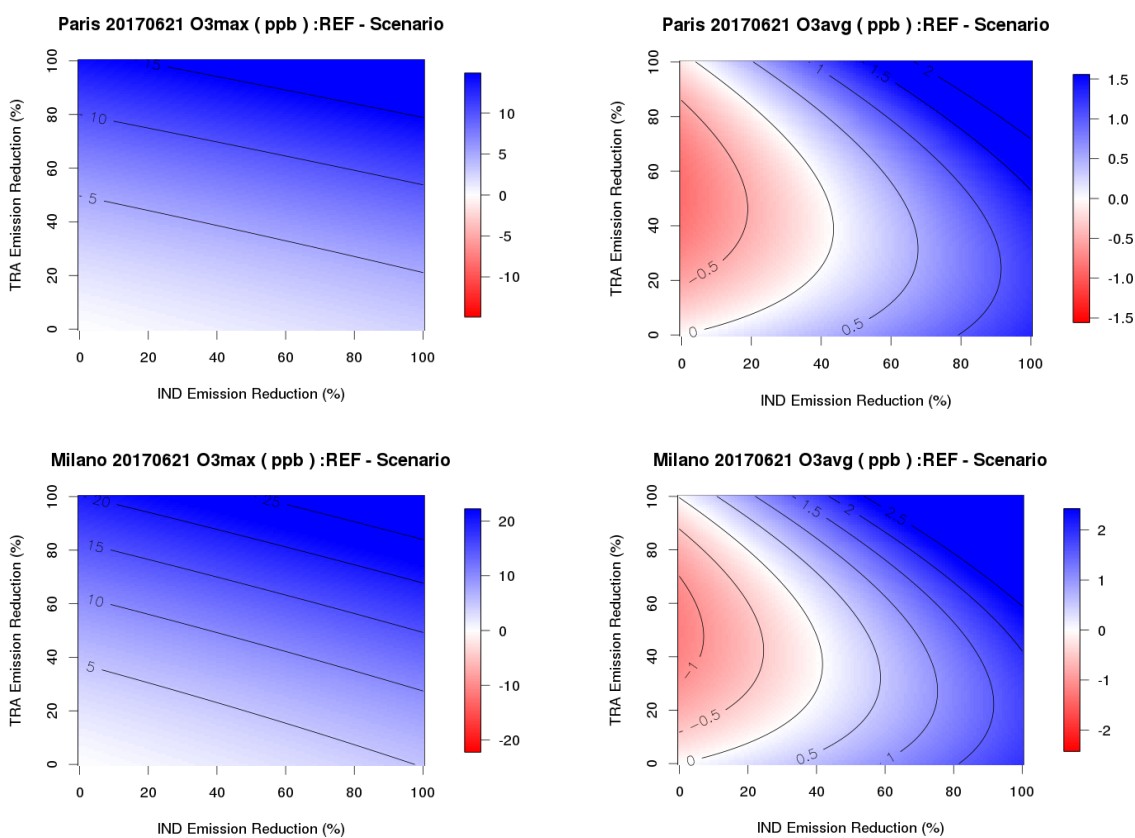

**Figure 17: Same as Figure 16 for ozone daily maximum (left) and ozone daily mean (right) and Brussels, Paris and Milano for a high ozone day (20170621).**