# Peer review of "Air Control Toolbox (ACT\_v1.0): a flexible surrogate model to explore mitigation scenarios in air quality forecasts."

_Geoscientific Model Development, 2020_

## Author Comment (AC1)

**Author reply to the reviewers of "Air Control Toolbox (ACT_v1.0): a machine**
**learning flexible surrogate model to explore mitigation scenarios in air quality**
**forecasts." Submitted to Geos. Model Dev. By Augustin Colette, Laurence**
**Rouïl, Frédérik Meleux, Vincent Lemaire, Blandine Raux**

We are deeply grateful to the three reviewers who provided constructive critics and excellent
reflexions to improve our manuscript, yet supporting the approach undertaken in this development.
We tried our best to address all concerns, with detailed answer in this document, and in track change
in the original manuscript.

In this response, the reviewer comments are highlighted in bold, author responses in plain text, and
modifications to the manuscript indented in track changes with page and line numbering referring to
the track change version of the revised manuscript.

**Community Comment #1 (https://doi.org/10.5194/gmd-2020-433-CC1)**

**ABTRACT. In the abstract, the authors should better explain the limitations of their approach.**
**Mainly, the fact that, in my understanding, scenarios are related to EU emission reductions, applying**
**constant sectoral reductions everywhere over the domain. These are quite important assumptions,**
**that limit the applicability of the tool to specific type of studies. Authors should better stress this**
**points.**

This comment is totally relevant. And the reviewer is right that is was not so clear in the abstract. We
have made it clearer with the following changes.

In the abstract: P1L26 of the revised manuscript

The selected approach makes ACT the first air quality surrogate model capable to capture non-linearities
in atmospheric chemistry response. Existing air quality surrogate models generally rely on a linearity
assumption over a given range of emission reductions, which often limits their applicability to annual
indicators. Such a structure makes ACT especially relevant to understand the main drivers of
air pollution  episode analysis. This feature is a strong
asset of this innovative tool which makes it also relevant for source apportionment and chemical regime
analysis. This breakthrough was only possible by assuming uniform and constant emission reductions for
the four targeted activity sectors. This version of the tool is therefore not suited to investigate short term
mitigation measures or spatially varying emission reductions.

**INTRODUCTION. The sentence "The two most widespread applications of atmospheric chemistry**
**modelling are (i) short term air quality forecasting, and (ii) long term analysis of mitigation strategies.**
**We introduce here the first toolbox able to address both issues at once," is for me misleading. As**
**long-term mitigation strategies is more for structural measures for which a forecasting model is not**
**useful. Or better, you can use a forecasting model, but it is more useful to check the change of**
**concentrations in the long-term, more than the day to day change. Furthermore, when designing**
**mitigation scenarios, is quite uncommon to have constant reductions over the whole domain. Please**
**better specify these limitations.**

A clarification of the fact that only long-term mitigation strategies are targeted has been added in the
first sentence of the introduction,  P2L12 of the revised manuscript

We introduce here the first toolbox able to address both issues at once, so that the user can explore the
benefit of a long term  emission reduction control strategy for the current-day air quality forecast.

The assumption that emission reductions are homogeneous and constant were already clearly
highlighted later in the introduction (P3L19-20 of the submitted manuscript). But we further expanded
that paragraph to highlight that we do not intend to "check the change of concentrations in the long-
term" as rightfully pointed out by the reviewer. It is indeed the day-to-day change which is our focus.
When it comes to decide on which of them is "more useful", we would leave it up to potential users.
The position of the authors is that it remains interesting to know the drivers of the daily variability in
air pollution. We can make an analogy in the field of climate science where some studies intend to
quantify the role of climate change in current weather extremes. This is indeed different than assessing
the benefit of long-term strategies to mitigate climate change. Understanding the difference is
important but deciding on which of them is more useful is a matter of personal opinion. Nevertheless,
the reviewer is right that such assumption must be well explained to the reader, and we hope that the
revision proposed below addresses this point.

In the introduction, P4L16 of the revised manuscript:

The only two simplifications limiting the range of application of ACT are that emission reductions are
assumed to apply (i) over the long term
and (ii) over the whole modelling domain uniformly. ACT is therefore not suited to assess the
benefit of emergency measures to mitigate air pollution episodes. It also not designed to assess the impact
on long term exposure to air pollution. The purpose of the tool is rather to assess the main activity sectors
driving the day to day air pollution variability. Several diagnostics are proposed to achieve this by
providing the source allocation and chemical regimes at various receptor locations, as well as mapping
the benefit of reducing emissions over the whole modelling domain. If we take the analogy with climate
change, the scope of ACT is analogous to attribution studies where one intends to quantify the role of
climate change in the current meteorological situation. Which is different than assessing the benefit of
greenhouse gas strategies to mitigate long term climate change.

A similar note was also added in the conclusion, P24L19 of the revised manuscript:

To our knowledge, this model is the first surrogate, or emulator, able to cover short time scale for air
pollution studies. This development was only made possible by assuming uniform and constant emission
reductions for the four targeted activity sectors. The fact that emission reductions are assumed to be
applied over the long term makes ACT not suited to assess the benefit of emergency measures to mitigate
air pollution episodes. The purpose of the tool is rather to assess the main activity sectors driving the day
to day air pollution variability.

**SECTION 3. The paper in section 3 is much too long and full of details. I would propose to restructure**
**the work, moving to Supplementary Material the technicalities of the work, and keeping only the**
**most relavant parts of the paper in the 'Main' part.**
Section 3 of the paper constitutes the main body of the design of the ACT tool. In the submitted
manuscript it is 7.5 pages long out of 22 (without references and figures) which can also be considered
an appropriate level of detail. We believe that a model description paper in GMD is well suited for such
a methodological description, unlike other journals where methods are moved to the supplementary
materials. But we can reconsider this position if the editor thinks fit.

 **Anonymous Referee #1 https://doi.org/10.5194/gmd-2020-433-RC1**

**Abstract and introduction: Overall, the paper is well written but the authors should introduce more**
**clearly the objective of their study and the general form of the tool they aimed to build. While**
**reading the paper, it took me some time to understand precisely which type of tool they were**
**developing. One reason may be that in the abstract and introduction, the authors are often talking**
**about a "surrogate model" to tackle the complexity of geophysical chemistry-transport models**
**(CTM). I was a bit confused at the beginning since this tends to suggest that they want to build a**
**CTM surrogate model (able to predict concentrations based on emissions) while actually they want**
**to develop an emission scenario surrogate model (able to predict the concentration changes in**
**response to emission changes). Although some sentences indeed refer to the later, some others refer**
**to the former, which maintains the confusion (e.g., P2L6 : "an emulator, or surrogate model, of a**
**comprehensive air quality model"). While reading the paper, we progressively understand what the**
**authors were referring to before, but the first part of the paper could be clarified so that things are**
**clear earlier in the text. I would personally understand better and easier the goal of their study by**
**introducing it in such a way :**
**The overall objective of the authors is to build a tool for conducting flexible emission scenario**
**analysis on an every-day basis, for exploring the impact of different mitigation strategies on the**
**current forecast. More specifically, they want the tool to estimate the impact of a given set of**
**emission reductions (in this case, looking at the options available for the user on the ACT web**
**interface, 21 emission scenarios of emission reduction bins of 5%, namely 0%, -5%, -10%..., -95%, -**
**100%) applied to a given set of emission sectors (in this case, 4 emission sectors). Simulating**
**explicitly all possible combinations would require a total of 21^4=194,481 emission scenario**
**simulations (or 101^4=104,060,401 simulations with emission reduction bins of 1%), which is**
**obviously not feasible for computational reasons. This motivates the need for developing, in each**
**cell of the domain and for each day of forecast, (1) a regression model able to estimate the pollutant**
**concentration responses to different emission reductions based on (2) a reasonable number of**
**emission scenario simulations that could be run in parallel to the reference CTM simulation on an**
**every-day basis.**
**Then, this naturally justifies the type of regression model chosen. Indeed, although more complex**
**regression models can eventually achieve better accuracy, they require larger training datasets (in**
**this case, more emission scenario simulations). As a compromise between these two aspects, this**
**motivates the choice of multivariate second-order polynomial models with interaction terms able to**
**capture some of the non-linearities existing between concentration changes and emission change**
**(at least in some emission sectors) while remaining simple enough, in the sense that only a few**
**coefficients need to be fitted, which allows relying on a limited number of emission scenario**
**simulations.**

We are sorry if the reviewer found it difficult to understand the scope of the work in the abstract and
introduction. We very much appreciate the suggestions for improvement, which helped in improving
the abstract and introduction.

In the abstract: P1L16 of the revised manuscript

As such we take the best of the physical and chemical complexity of CTMs, operated on high performance
computers for the everyday forecast, but we approximate a simplified response function that can be
operated through a website to emulate the  sensitivies of the atmospheric system to anthropogenic
emission changes for a given day and location.

In the introduction: P3L22 of the revised manuscript

The overall objective is to offer the users a high degree  of flexibility  to explore any  mitigation scenario through a web interface. We are targeting four activity sectors and for each of them the available emission reduction should cover the whole 0 to 100% range, for instance by 5% increments. Using an explicit approach, this would imply $21^4$ or 194,481 chemistry transport simulations which is obviously not feasible for computational reasons. The whole point of the present paper is therefore to design a methodology to reduce this number down to a reasonable number of simulations which can be performed on a daily basis and used as a training set to calibrate a simple response model.

The selected architecture of the response model is a polynomial function whose coefficients can be explicitly estimated from a limited number of simulations. Fitting more complex regression models could achieve better accuracy, but only at the cost of a larger training set. We will see that a quadrivariate (four activity sector) second order polynomial can be well constrained with only a dozen of chemistry transport simulations, which is quite acceptable with regards to the operational constrains of every-day production in an air quality forecasting system.

With this approach, all the complexity of the atmospheric chemistry response to incremental emission changes is embedded in  a polynomial surrogate which is derived  every day from  on a subset  of chemistry-transport simulations.

The use of a complete chemistry-transport model to build the training simulations allows  accounts for all the important processes bearing upon the forecasted air quality, including long-range transport and chemistry. And the surrogate model is able capture some of the non-linearities existing between concentration and emission changes while remaining simple enough, in the sense that only a few coefficients need to be resolved.

**Section3 : I agree with the first reviewer that section 3 could be shorter. Also, it is not very clear to which extent the step-by-step approach - (1) univariate first/second/third order model, (2) bivariate second order model with interaction terms, (3) quadrivariate first/second order model with interaction terms - is followed only for pedagogical purposes or if (at least part of) the methodology is really sequential (it seems so). Please clarify.**

As replied to CC1, we regret this impression of a too long section 3 (which is only 7.5p out of 22). We leave it up for the editor guidance to move part of it to the supplementary material, but we also understood that it is the very scope of GMD to publish such details, while other journals require to keep the methodological details in the annex.

This sequential approach has indeed some pedagogical advantages. But the reviewer is right in pointing out that the design of the method indeed follows this series of steps. That is precisely why we kept this structure in the paper as we believe that the same sequence should be followed when implementing the method in a different context. The ACT tool presented here covers Europe at 0.25 degree resolution. But one may want to replicate the approach in other areas, or at finer spatial resolution. Nothing guarantees that the same model structure would hold in such cases. And we advise that the same series of steps are followed.

Section 3.3 P15L4 of the revised manuscript

The methodology followed in Sections **Erreur ! Source du renvoi introuvable.** and **Erreur ! Source du renvoi introuvable.** consists in selecting first the optimal structure for univariate models, before investigating bivariate models including interactions terms. Such a step-by-step approach allows a clear introduction of the methodology. This presentation has a clear pedagogical advantage. It is also relevant for a potential development of a similar approach in a different context. One could consider developing an ACT tool over a different region, or at higher spatial resolution. But, in that case, nothing guarantees that the same model structure would be selected, in particular if non-linearities affect different activity sectors.

However, such a sequential approach carries a risk of not selecting the optimal structure, as pointed out for stepwise regression approaches. Indeed, with a sequential method, we assume that the optimal structure and training scenario remains valid when including interactions whereas there is a possibility that the addition of an interaction term could change the selection of univariate terms.

**Section 3: If I understand correctly, pairs of emission sectors with most important interactions are chosen in section 3.2 based on different two-variable polynomial regression models, but how this ensures that it remains then the most appropriate choice in the four-variable polynomial models developed in section 3.3? The authors acknowledge that "by doing [such a step-by-step approach], [they] assume that the optimal structure and training scenario remains valid when including interactions whereas there is a possibility that the addition of an interaction term could change the selection of univariate terms." However, in the type of model proposed in section 3.3, the authors keep only the interaction terms selected before. Also, to test this final model, they retain only the subset of emission scenario selected before. These different methodological aspects should be clarified.**

Despite interesting from a pedagogical point of view, the chosen sequential approach carries a risk of deviation in the optimum also identified for stepwise regression. That is why, after having identified an optimum with a sequential method, we explore again all possible combinations in section 3. With this systematic combination of all models, we show that the model selected with the sequential approach turns out to be the optimum for ozone, but not for $PM_{10}$ (by only a small margin). All the relevant pairs of interaction terms are kept. Only those involving the RH sector are excluding because it has been demonstrated to be well approximated with a linear fit.

Section 3.3 P15L23 of the revised manuscript

Such a model requires two training scenarios for AGR, IND, TRA, one for RH and one for each of the three interaction terms. With this approach all possible 2-term interactions are indeed taken into account as only those involving RH are excluded because we have demonstrated earlier that this factor could be well approximated with a linear relationship (and therefore irrelevant for second order interactions).

**I wonder to which extent not considering any emission scenario with reductions applied to 3 or 4 sectors simultaneously (besides the scenario with all 4 sectors reduced by 100%), for the training or at least the testing is a strong limitation in this study. This should be discussed (and ideally tested).**

This comment is very relevant, and one could indeed consider further increasing the refinement in the surrogate model. The only reason not to engage in that direction is a trade-off between computational cost of the training scenario and performances of the surrogate.

A discussion has been added Section 3.4 P17L7 of the revised manuscript:

Further increasing the degree of the polynomial would certainly improve the quality of the surrogate model. The only two reasons not to engage in that direction are: (i) avoid increasing the computation burden with more training scenario and (ii) considering that the performance achieved at order 2 are already very satisfactory. The only higher order interaction scenario removing emissions from all 4 sectors is designed as a closure to avoid any potential negative concentrations.

**Additional comments :**

**P1L13 : "flexibility" : please be more specific on which type of flexibility you are talking about (here, flexibility on the emission forcing)**

Abstract P1L13 of the revised manuscript:

However, their complexity prohibits offering a high level of flexibility in the tested emission reductions.

**P4L11 : "Note however that those periods do not constitute a specific training period for the surrogate model which is intended to be re-fitted to new CTM simulations every forecasted day in an automated machine learning approach." Do the authors mean that all regression models in this study are trained only based on the current forecast day? Please clarify. In total for this study, the authors have performed 46 CHIMERE simulations for each of the 4 months mentioned in Section 2.1, is it correct?**

A clarification has been added in Section 2.1, P5L17 of the revised manuscript:

Note however that those periods do not constitute a specific training period for the surrogate model which is intended to be re-fitted to new CTM simulations every forecasted day in an automated machine learning approach. Those episodes are only selected to identify the best design for the surrogate and assess its performances in reproducing the full CTM. We will see in Section **Erreur ! Source du renvoi introuvable.** that in total 46 simulations were required to identify the optimum surrogate model structure and demonstrate its performances. All these 46 simulations covered the 4 months selected in the years 2015 to 2017 to capture a variety of air pollution episodes. But once the structure of the surrogate model is identified, the model itself is intended to be calibrated automatically on the basis of the day to day forecast.

**P4L25 : "operational analyses of the IFS" : not IFS forecasts?**

Indeed for this development phase, IFS analyses were used, but in operational setup, ACT is fitted on CHIMERE simulations driven by IFS forecasts

**P5L20 : "we conclude that the surrogate model will be at most a third order polynomial, less if interaction terms are accounted for." Please clarify that this is motivated by the number of coefficients to be fitted.**

Section 2.3, P7L1 of the revised manuscript:

Considering the goal to cover four activity sectors, and the operational constrain to compute only with about 10 to 15 training scenarios every day, we can derive a limit in the number of coefficients which we can estimate. We conclude that the surrogate model can will be at most a third order polynomial, or even less if interaction terms are accounted for.

**P6L4 : "emissions are applied uniformly over Europe" should be "emission reductions…"**

Section 2.4, P7L14 of the revised manuscript:

In all cases, emissions reductions are applied uniformly over Europe and for all chemical species.

**P7L16 : "both dates" : which dates are you referring to?**

Section 3.1.1, P9L2 of the revised manuscript:

The residential heating (RH) sector contributes mainly with primary $PM_{10}$ emissions or organic aerosol precursors, and its response is the closest to linearity for 20th March 2015 and 1st December 2016 both dates and the three selected cities.

**P9L1 : "testing and validation" should be "training and testing"**

Section 3.1.1, P11L1 of the revised manuscript:

The corresponding list of scenarios available for training and testing and validation are summarized in **Erreur ! Source du renvoi introuvable.**, there are 5, 10, and 10 combinations for the linear, quadratic and cubic forms, respectively.

**FIG3 : Please describe in the legend the meaning of the numbers into brackets (I guess they**
**correspond to the minimum and maximum error over the domain?)**

The legend of Figure 3 has been changed following a comment from Anonymous Reviewer #2.

**P10L1 : "perform" should be "performs"; "largest emission reduction" should be "largest emission**
**reduction"**

The first change was taken into account. I could not find the issue with the second point

Section 3.1.2, P11L24 of the revised manuscript:

The third order polynomial performs best, except the last model which uses only the largest emission
reduction and is therefore too weakly constrained for the lower range of reductions

**P11L6 : "yield" should be "yields"**

Section 3.1.2, P13L4 of the revised manuscript:

On the contrary, for Industry and Traffic, we opt for a quadratic model (using 60% and 100% reductions,
index 14 in the x-axis of **Erreur ! Source du renvoi introuvable.**) which yields median errors below
0.1% (0.099 and 0.028%, for IND and TRA, respectively) and the gain in term of median error is a factor
3-4 compared to the linear forms.

**Sect. 3.1 : Results are discussed for PM10 and briefly mentioned for O3max but not for the other**
**pollutants**

This is because the same reasoning applies for other pollutants and metrics, for which only overall
performances are presented in Section 3.4

Section 3.4, P16L27 of the revised manuscript:

More specifically, the single highest error over the 864 248 grid points and days considered is 7.5%, 7.9%,
4.8%, 2.8% and 2.9% for $PM_{10}$, $PM_{2.5}$, $NO_2$, $O_3$max, $O_3$avg<s>the</s>

**Table 2 : a comma is missing between IND30 and IND60**

Added in Table 2

**P12L10 : It seems to me the authors are actually talking about lines 3 and 4 where no interactions**
**are included in the training set, please correct if needed.**

Section 3.2, P14L28 of the revised manuscript:

First a bivariate quadratic model without interactions is trained with the 30% and 60% reduction levels
and tested against corresponding interaction scenarios. Taking the example of agriculture & industry, we
would have the two training and testing configurations in lines 3<s>1</s> and 4<s>2</s> of **Erreur ! Source du renvoi**
**introuvable.**.

**All formulas : In order to make formulas easier to read (especially the longest ones), I suggest to**
**introduce another variable such as DELTA^sector=EPSILON^sector-EPSILON^ref.**

This notation was introduced in all relevant equations in Section 3.1.2, 3.2, and 3.3 (P10L10, P14L1,
P15L15 of the revised manuscript)

$$C_{i,j} - C_{i,j}^{ref} = \alpha_{i,j} \cdot \left( \delta_{i,j}^{sec} \cancel{\varepsilon_{i,j} - \varepsilon_{i,j}^{ref}} \right) + \beta_{i,j} \cdot \left( \delta_{i,j}^{sec} \cancel{\varepsilon_{i,j} - \varepsilon_{i,j}^{ref}} \right)^2 + \gamma_{i,j} \cdot \left( \delta_{i,j}^{sec} \cancel{\varepsilon_{i,j} - \varepsilon_{i,j}^{ref}} \right)^3$$

$$C^{agr,ind} - C^{ref} = \alpha^{agr} \cdot \delta^{agr} + \beta^{agr} \cdot (\delta^{agr})^2 + \alpha^{ind} \cdot (\delta^{ind}) + \beta^{ind} \cdot (\delta^{ind})^2 + \gamma \cdot (\delta^{agr}) \cdot (\delta^{ind})$$

$$C^{agr,ind,rh,tra} - C^{ref}$$
$$= \alpha^{agr} \cdot (\delta^{agr}) + \beta^{agr} \cdot (\delta^{agr})^2 + \alpha^{ind} \cdot (\delta^{ind}) + \beta^{ind} \cdot (\delta^{ind})^2 + \alpha^{rh} \cdot (\delta^{rh}) + \alpha^{tra}$$
$$\cdot (\delta^{tra}) + \beta^{tra} \cdot (\delta^{tra})^2 + \gamma^{agr,ind} \cdot (\delta^{agr}) \cdot (\delta^{ind}) + \gamma^{tra,agr} \cdot (\delta^{tra}) \cdot (\delta^{agr})$$
$$+ \gamma^{tra,ind} \cdot (\delta^{tra}) \cdot (\delta^{ind})$$

**P14L13 : "This final model is tested the" should be "This final model is tested against the"**

Section 3.4, P16L23 of the revised manuscript:

> This final model is tested against the 34 available scenarios not used in the training.

**P17L6 : "ammonia emissions December is" should be "ammonia emissions in December are"**

Section 4.2.2, P19L28 of the revised manuscript:

> Given the strong seasonality of fertilizer spreading, ammonia emissions in December are  likely due mainly to livestock emissions.

**P18L4 : "the allocation we display here" : where?**

Section 5, P20L24 of the revised manuscript:

> Unlike source apportionment, the allocation we introduce  here indeed shows the reduction of concentration that can be achieved by removing totally the emissions of an activity sectors.

**P18L14 : "assess" should be "assessed"**

Section 5, P21L7 of the revised manuscript:

> Here, the contribution of each activity sector is assessed by setting its emissions to zero (referred to as top-down brute force method in (Clappier et al., 2017a)).

**ACT web interface : Why only O3 is proposed and not O3max or O3avg? Also, the source allocation model is useful for analysing pollution episodes and I would suggest to include it on the ACT web interface in the future.**

Adding O3avg as well as source allocation and chemical regimes are currently being considered for inclusion in the web tool. Given that they are also sometimes more difficult to understand for a non-specialised audience, the publication of the present paper was an important prerequisite prior to such a public dissemination.

**Anonymous Referee #2 https://doi.org/10.5194/gmd-2020-433-RC2**

This paper addresses an important topic, is interesting and well written. The authors develop a surrogate model for a full CTM to investigate ambient pollution changes in response to sectoral emission reductions. Currently the application is limited to episodes and uniform emission changes over Europe which, as also remarked by the other anonymous referee, limits the practical applicability for policy analysis, but the authors also emphasize that they are planning to extend the tool in this direction. It will be interesting to follow this development.

I have a few comments which should be addressed, but these are mostly clarifications and should not require major changes before publication.

General comments

**1. I confess I am not an expert for machine learning but I was wondering whether the title is really appropriate. The toolbox as described in this paper uses a parametric regression (first and second order with interaction terms) to approximate the behaviour of a full CTM, and while I think the methodology is sound, I'm not sure if this can be really termed machine learning**

The reviewer is right that we used a probably too broad definition of machine learning. The toolbox is designed to be calibrated automatically, but the actual structure of the surrogate is designed manually and indeed relies on a parametric model. In using that term in the paper, we followed a general tendency to include any statistical methods under the terminology of machine learning, which is perhaps not justified, so that we removed these instances in the title (P1L1), abstract (P1L14), and section 2.1 (P5L15).

**2. I struggled a bit to understand how the regression is set up. When looking at Eq 1, one could get the impression that everything is known, and the equation can simply be solved with a unique solution in each grid cell for the parameters alpha, beta, gamma (if Ci,j and ε are scalar values averaged for the time period of interest) and no regression is needed if the right number of training scenarios is used (2 for a linear fit, 3 for quadratic etc). Is this what is done, or is there a regression involved in the sense of solving an overdetermined equation system and estimating optimal alpha, beta, gammas? Also, should Eq 1 not have a sectoral dimension index in alpha, beta, gamma and epsilons?**

We apologize that the term "regression" was misleading. The system is not overdetermined: because of computational constrains we don't have the luxury to run unnecessary simulations with the full chemistry transport model.

This has been made clearer by avoiding referring to "fit" or "regression" in more than 20 instances throughout the paper.

The sectoral dimension was lacking in Eq 1, it has now been added, P10L10 of the revised manuscript:

$$C_{i,j} - C_{i,j}^{ref} = \alpha_{i,j} \cdot \left( \delta_{i,j}^{sec} \varepsilon_{i,j} - \varepsilon_{i,j}^{ref} \right) + \beta_{i,j} \cdot \left( \delta_{i,j}^{sec} \varepsilon_{i,j} - \varepsilon_{i,j}^{ref} \right)^2 + \gamma_{i,j} \cdot \left( \delta_{i,j}^{sec} \varepsilon_{i,j} - \varepsilon_{i,j}^{ref} \right)^3$$

**3. Besides the limitation that emission reductions which can be explored with this tool are uniform across Europe, also they are specified as reduction of emissions by sector since the coefficients are expressed by sector and not by pollutant, meaning that all pollutants emitted from this sector are reduced proportionally. In practice, emission control policies may reduce different pollutants from**

**the same SNAP sector to a different extent, which cannot be simulated with the current setup. It would be good to mention this somewhere.**

We fully agree with this comment, but would like to emphasize that this is complementary with other approaches, but not a limitation per se. Otherwise one should consider that surrogate models targeting individual species are also limited in the fact that they do not provide the flexibility to assess activity sectors. The following sentences were added P7L14 of the revised manuscript:

> In all cases, emissions reductions are applied uniformly over Europe. It is also important to stress that  all chemical species for a given activity sector are reduced by the same amount. This is different and complementary with the approach chosen for instance in the EMEP source receptor matrices (Amann et al., 2008) or the SHERPA tool (Pisoni et al., 2017).

**P 5, L 6: The number of training simulations should be constrained by more fundamental principles than the availability of computing time?**

The tool is specifically designed to be implemented in an operational daily air quality forecasting system. Therefore computing time constrains are indeed a critical limitation to be taken into account in its design.

**P 7 – I would emphasise a bit more the complete reverse response for TRA reductions wrt O3 avg vs peaks. In this context it would be good to mention what is assumed regarding NO/NOx ratio in industrial vs traffic emissions, this may play a role here.**

This paragraph has been expanded to emphasise this difference and include a mention to the role of NO/NOx ratios, cf P9L17 of the revised manuscript:

> This very distinct sensitivity of $O_3$ daily maximum and daily averages is due to the titration process, which affects more strongly night-time, low, ozone concentrations and is therefore less visible on the daily maximum. In Figure 2, it is a specific feature of the traffic sector, which is not seen for instance for industrial emissions. It should be noted that the $NO_2$/NOx ratio is very different for both sectors. In the CHIMERE modelling setup used here, 20% of NOx emissions are allocated to $NO_2$ for the traffic sector, while for other sources including industry it is only 4.5% of NOx emissions that are constituted of $NO_2$. But this larger share of NOx emissions as NO in the industry sector would rather imply a lower titration of $O_3$ by NO in traffic emissions compared to industrial sources. And in summer, NO and $NO_2$ will even out very fast at the spatial scales of a regional chemistry transport model. The stronger impact of titration when reducing traffic emissions in Figure 2 is therefore rather a consequence of the overall larger share of total NOx emissions compared to other activity sectors.

**P10 L 22: I am confused what is actually calculated and shown here. Maybe I'm too long after my last statistics class but I don't exactly understand why a confidence interval is calculated here (for what estimated quantity?) and where the formula comes from. Wouldn't it be more useful to show just the 95th or 97.5th percentile of this error distribution?**

We used the default implementation of boxplot displays in R, and we appreciate this comment which provided us the opportunity to fix an inaccuracy in the paper. After a closer look it appears that these whiskers are not providing the confidence interval but rather 1.5 the interquartile range. See new formulation in Section 3.1.2, P12L19 of the revised paper:

> hiskers extend to 1.5 times the interquartile range from the borders of the box, and the extreme points lying outside of that range are  also provided.

**Section 3.4: Is the final model setup that the authors arrive at in Section 3.4 stable or is this re-evaluated for new time periods in the operational tool?**

This has been clarified in 3.4, P17L1 of the revised paper:

> This structure is selected because of is good performances for all relevant ambient air pollutant as demonstrated here over a range of air pollution episodes. In the operational model, the overall structure of the model is frozen, while only the coefficients of the surrogate are recomputed every day on the basis of the current air quality forecast. It should be noted however that any evolution of the model such as increasing the spatial resolution, implementation over a different geographic area, or including other activity sector would require revising this structure.

**Section 4.2.2: The conclusion that the December episode is mainly driven by agricultural emissions is rather unexpected, I would have expected a higher contribution from residential heating. How well does the CHIMERE model in this configuration perform in reproducing the measured concentrations of these episodes? The authors note that the emission inventory used may underestimate residential sector emissions, which is an important caveat to highlight when drawing conclusions for wintertime episodes.**

We agree that it is important to highlight the limitations introduced by the underlying model and data. This has been further emphasised in Section 5 on Source Apportionment, P21L17 of the revised manuscript:

> This underestimation of residential emission has been widely documented as it has a very strong impact on model performances to capture wintertime peaks (Denier van der Gon et al., 2015). The expected future improvements in the consistency of reported condensable PM in emission inventories should improve notably model performances. Until then, this feature provides us an opportunity to highlight that the diagnostics derived with the surrogate model remains highly sensitive to underlying hypotheses in emission inventories and chemistry transport modelling.

All the following technical corrections were implemented, as noted below:

**P 6, L 5. "Emissions": emission changes / reductions**
Done

**P9 L21 performance (singular)**
Done

**P10 L12: in order to reduce the number of performances in this sentence, I suggest to replace the middle one with 'done'**
Done

**Fig 3: The legend is messy and the line colors are not distinguishable. I suggest to change the figure layout. For example, unify to one table with scenario numbers like in Fig 4.**
Figure 4 has been modified to account for this suggestion. The same modification was applied to Figures S.3, S.4, and S.5.

**Fig 4: color scales are exhausted for some of the figures**
This is unfortunately true, but we selected the color scale range specifically to provide a relevant level of details for the quadratic models which provide the best results

**P13 L19 – "We are left already" -> We are still left**
Done

**Figure 7 caption: raw -> row**
Done

**P14 L13 – is tested with / against**

Done

**Figure 11 caption: 2015903 ?**
Done

**Figure 12 top right - caption unclear**
A clarification was added in the main text introducing that figure (Section 4.2.1, P18L14 of the revised manuscript)

**P16 L 10: NOx from industry and heating**
Done

**P16 L18: in Western Europe**
Done

---

## Author Response (AR2)

**Author reply to the reviewers of "Air Control Toolbox (ACT_v1.0): a flexible surrogate model to explore mitigation scenarios in air quality forecasts." Submitted to Geos. Model Dev. By Augustin Colette, Laurence Rouïl, Frédérik Meleux, Vincent Lemaire, Blandine Raux**

In this response, the reviewer comments are highlighted in bold, author responses in plain text, and modifications to the manuscript indented in track changes with page and line numbering referring to the track change version of the revised manuscript.

**The authors have done a good job in responding to many comments, and have improved the paper. I have only two remaining issues which prevent me from recommending publication of the manuscript as-is. The first comment relating to the model formulation is substantial but hopefully only due to unclear notation:**

**1) Eq 1 on p10: I appreciate that the authors have made an effort to explain the methodology better. However, on reading this part again, I am still quite confused:**

**a) Where is the source pollutant? I guess alpha, beta, gamma, deltas and epsilons are actually source pollutant specific? This should be clarified. The 'pollutants of interest' mentioned in line 4 (and which the C in Eq 1 refer to) are only the ambient pollutants, resulting from emissions of several precursor species.**

**b) The concentration changes in one grid cell (i,j) are related here to the emission changes \delta^{sec}_{i,j} of sector sec in this same grid cell only. I assume this is only an error in the notation, otherwise I don't understand how this can sensibly work since ambient concentration changes in one grid cell are necessarily related to emission changes elsewhere: This formulation would completely ignore any transport of pollution, which is both unrealistic as well as in contradiction to the authors' own statements in the introduction (p4, l 9-10). In particular, with this formulation I would not see how any effect of agricultural emission (changes) can be found in Paris in the March episode, since there are certainly no agricultural NH3 emissions in the grid cell located in central Paris. I assume these are total sectoral emission changes, i.e. \delta^sec = \sum_{i',j'} \delta^sec_{i',j'}.**

We are very grateful for the relevant remark pointing an inaccuracy of the notation in the manuscript. This has now been revised as highlighted below. To answer directly here the questions of the reviewer:

(a) alpha, beta, gamma are indeed ambient air pollutant specific and also varying in space. Delta are activity sector specific, but uniform in space and also applied identically to all emitted species of a given activity sector.

(b) it is right that spatial indices (I,j) should not have included for the emission reduction factor, which is applied uniformly over the whole domain.

The proposed modification is P10 L5 of the revised manuscript:

> For each day, and each pollutant, a polynomial model is calibrated at each grid point of the modelling domain. We introduce the following notations for a third order polynomial, with $\alpha_{i,j}, \beta_{i,j}, \gamma_{i,j}$ the coefficients (the later two being nullified for linear or quadratic forms):

$$C_{i,j} - C_{i,j}^{ref} = \alpha_{i,j} \cdot \left(\delta_{i,j}^{sec}\right) + \beta_{i,j} \cdot \left(\delta_{i,j}^{sec}\right)^2 + \gamma_{i,j} \cdot \left(\delta_{i,j}^{sec}\right)^3$$

Where:

$$\delta_{i,j}^{sec} = \left( \frac{\varepsilon^{sec}}{\varepsilon^{ref}} \varepsilon_{i,j}^{sec} - \varepsilon_{i,j}^{ref} \right)$$

- $C_{i,j}^{ref}$ is the air pollutant concentration (for either PM$_{10}$, PM$_{2.5}$, O$_{3max}$, O$_{3avg}$, or NO$_2$) modelled with the CTM for the reference simulation with emissions $\varepsilon_{i,j}^{ref}$

- $C_{i,j}$ is the air pollutant concentration modelled with the CTM for the sensitivity simulation with reduced emissions for sector "sec": $\varepsilon_{i,j}^{sec}$ reduced by a uniform factor $\delta^{sec}$ over the domain. In addition to being uniform in space, the reduction factor is also identical for all emitted precursor species since it is applied to the whole activity sector.

- throughout the paper, the coefficients α, β, and γ of such polynomials will be computed for each *i,j* pair of latitude, longitudes indices in the geographical modelling domain, so that the indices will be dropped in the following notations.

**2) The lines in Fig. 3 are still not distinguishable. This could easily be fixed by iterating through colors and line styles independently, i.e. one color series with solid lines, one with dashed, one with dotted.**

The color scheme was changed according to this suggestion for Figure 3 as well as Figures S.3, S.4 and S.5.